# Sensors Made of Natural Renewable Materials: Efficiency, Recyclability or Biodegradability—The Green Electronics

**DOI:** 10.3390/s20205898

**Published:** 2020-10-19

**Authors:** Benoît Piro, Hoang Vinh Tran, Vu Thi Thu

**Affiliations:** 1ITODYS, CNRS, Université de Paris, F-75006 Paris, France; 2School of Chemical Engineering, Hanoi University of Science and Technology (HUST), 1st Dai Co Viet Road, 10000 Hanoi, Vietnam; hoang.tranvinh@hust.edu.vn; 3Vietnam Academy of Science and Technology (VAST), University of Science and Technology of Hanoi (USTH), 18 Hoang Quoc Viet, Cau Giay, 10000 Hanoi, Vietnam; vu-thi.thu@usth.edu.vn

**Keywords:** sensors, degradable, recyclable, renewable, electronics

## Abstract

Nowadays, sensor devices are developing fast. It is therefore critical, at a time when the availability and recyclability of materials are, along with acceptability from the consumers, among the most important criteria used by industrials before pushing a device to market, to review the most recent advances related to functional electronic materials, substrates or packaging materials with natural origins and/or presenting good recyclability. This review proposes, in the first section, passive materials used as substrates, supporting matrixes or packaging, whether organic or inorganic, then active materials such as conductors or semiconductors. The last section is dedicated to the review of pertinent sensors and devices integrated in sensors, along with their fabrication methods.

## 1. Introduction

Since the first small electronic appliances developed for the broad public in the late 1970s, sales of electronic equipments have increased at a rate which was not expected even fifteen years ago. After quartz watches in the 1970s and 1980s, portable personal computers in the 1990s and the beginning of the 2000s, smartphones are certainly now the most widespread devices. This is probably just the beginning of the invasion of our everyday life by smaller and smaller, smarter and smarter devices linked by the Internet of Thing (IoT), to monitor our health, the contents of our fridges and settings on our central heater at a distance, or to control our keyless doors. These appliances, which sometimes look ineffective or useless in 2020, will probably be seen as essential in 2030. With this exponential development comes an environmental problem, however, not only due to the possible increase in energy demand in order to run all these devices, but also due to the accelerated obsolescence of this equipment, causing a huge fast-growing demand for rare materials along with a no less huge source of waste, which is becoming a serious pollution problem. See, for instance, Keedee et al. [1] for an overview of these problems. In Europe, the 2018 objectives for small household devices, electronic tools or even medical devices were 75% collection and 55% recycling (in terms of number of collected objects, not in terms of weight content for each object) [2], which is dramatically poor but, considering the way garbage is actually collected, cannot be much higher in practice. Several solutions are available to better manage this waste: a more efficient collection of obsolete devices, a more efficient recycling, which is different, a slower turn-over (which is highly improbable), less precious materials in each device or the replacement of these actual precious materials (precious metals, rare earth, etc.) by natural or at least renewable materials, possibly biodegradable. 

Bauer et al. [3] published, in 2014, a short perspective article about disposable electronics where they gave their vision of the future of such devices, with a focus not only on organic or natural materials but also on transient metals and their oxides, particularly pertinent for making dissolvable batteries, supercapacitors, or simply conductive tracks. One of the strongest conclusions of these authors is that the IoT will not really develop before technical solutions are found concerning disposability. Mühl et al., in the same year [4], even developed the opinion that such degradable or biodegradable devices possibly never emerge commercially and remain as objects of curiosity in labs (they focused on bio-organic electronics and bio-inspired organic materials only, however, which is obviously more challenging than developing only (bio)degradable materials).

When focusing on materials rather than on devices, the reader can refer to several reviews which have been published during the last decade. For example, in 2010, Irimia-Vladu et al. were among the first to publish a review dealing with biodegradable materials for organic electronics [5], which were not only based on traditional silicon technology but also inspired from materials from nature. They published a second review in 2012 [6], dealing with biodegradable organic electronic materials, including substrates, dielectrics, semiconductors, conductors, and encapsulation materials, actualized in 2014 [7]. 

For this review, we considered publications whatever their number of citations, date of publication or the impact factor of the journal in which they were published. If a few publications started to deal with the biodegradability of sensors in the very beginning of the 2000s, it is a fact that a very large majority of the works in this field were published in the last five years. We tried to cite these works chronologicaly, in each respective section, to make evident the progress made. We organized the review as follows: after giving the expected outcomes of the domain and some definitions, the first section is dedicated to materials used as substrates, supporting matrixes or packaging (i.e., protective layers deposited directly on a component), classified as inorganic, organic or hybrid. Then, active materials (semiconductors, conductors, dielectric, etc.) are discussed and classified similarly. Then, without distinction between substrates, passive or active materials, some pertinent sensors and devices constituting sensors are reviewed. Biofuel cells used as sensors or utilized in power sensors are also discussed. Some fabrications methods generally utilized for making biodegradable or bioresorbable sensors are also reviewed. Recycling methods and green synthesis are occasionaly introduced but not developed here; to go further, the reader can, for example, refer to the review of Ponnamma et al. [8], published in 2019, which dealt with eco-friendly synthesized polymer materials.

## 2. Discussion

In this first section, technical terms will be defined and the expected outcomes of the domain exposed.

### 2.1. Expected Outcomes

#### 2.1.1. Definitions

Let us consider the different kinds of sensor which will probably invade our everyday life within the coming years, and which will need to have a limited ecological impact and be efficiently recycled. They are all individual constituents of common electronics, i.e., conductive tracks, resistors, capacitors, transistors, inductors and antennas, but also the batteries, the substrates used to place these elements and the packaging materials used to isolate and protect these elements. Package, substrate and electronic elements should be all degradable or, at least, designed to be easily separable during the recycling process. For the (bio)chemical sensors, the active sensing layer is also expected to be made of recyclable or degradable individual elements: nanomaterials, catalysts, membranes, etc. Materials will be discussed in Section 2.2 and Section 2.3 of this review. Their application in active elements is expected to be used in physical sensors (which already benefit from significant advances in the matter of degradability) such as pressure, strain, deformation, temperature, humidity, breath, heartbeat, and chemical sensors, which are much less developed for this purpose, as discussed in Section 2.4. 

Before going further, it is necessary to define some terms, such as “renewable” and “biodegradable”. On the one hand, a renewable material is a material which will, by natural processes, be replaced in nature after being consumed. This implies that the replacement process occurs at a reasonably fast rate compared to that of its consumption. On the other hand, a degradable material is something which will eventually break down to its basic components under defined physico-chemical conditions such as temperature, pressure, pH and, if biodegradable, upon the presence of microorganisms. Ideally, the basic degradation products are not toxic; however, this is not obvious and not general, and some materials degrade into harmful sub-products. A (bio)degradable material is expected to break down in a relatively short period of time, even if that particular (but important) parameter is often not explicitly given. This is particularly true for some polymers, for which (bio)degradation sometimes needs harsh conditions. For example, polylactic acid (PLA), which is a popular polymer routinely used in 3D printers and currently described as biodegradable, only degrades over 60 °C and in a humidity-saturated environment. In other words, if left behind in nature, it will not degrade. True degradability is therefore a point of vigilance. Among truly biodegradable materials, one finds paper and all wood derivatives (providing that they have not been blended with non-biodegradable materials and pre-treated with toxic chemicals, which is the common case), easily oxidizable metals or carbon derivatives, which are reviewed in detail in Section 2.2 and Section 2.3.

Besides the need for sustainable electronics, which implies recyclability and/or biodegradability, there are also bioresorbable materials. The two should not be confused, in the sense that bioresorbable materials are expected to be fully degraded (or dissolved) in vivo, without any toxic degradation product and under mild conditions (neutral pH, 37 °C, ambient pressure). In this article, bioresorbable materials and sensors will not be reviewed. To bioresorbable implanted electronics, more severe specifications should be applied, because materials composing a bioresorbable component must produce biocompatible degradation byproducts that can be rapidly metabolised. For more details on bioresorbable devices, the reader may refer to the review by La Mattina et al., published in 2020 [9].

Table 1 summarizes the various acronyms used in this review.

#### 2.1.2. Outcomes

##### Materials

Implanted bioresorbable electronics are expected to work in physiological conditions for a given amount of time only, and then be eliminated without the need for surgery. Before the sensor era, this property was expected to result from passive prosthesis only, typically bioresorbable endovascular stents [10] or bone reconstruction scaffolds [11]. Today, the landscape has completely changed, with the development of a variety of invasive sensors for personalized healthcare (glycaemia, hypoxia, pH, cancer biomarkers, inflammations, sepsis, etc.) [12,13] for which the need for bioresorbability is also coupled with the need to avoid an accumulation of electronic waste in the environment. To bypass difficulties associated with the bioresorbtion of invasive sensors, one strategy is to make sensors less invasive. This is the case of sensors in the form of dissolvable tattoos, which are developing rapidly today [14], or other kinds of external, wearable sensors, a few examples of which will be given throughout this review. See, for example, Hwang et al. in Section 2.2.2 and Kang et al. in Section 2.3.1.

For both biodegradable and bioresorbable materials, (bio)degradation is really effective when materials are engineered specially for that purpose. For the particular case of bioresorbable materials, it could even be advantageous to trigger the decay through changes in the physicochemical environment (pH, temperature, light, etc.) of the device, for a kind of *on demand* dissolution. It is therefore important to improve our knowledge of this matter. The reader can refer to the works from J.A. Rogers and coworkers [15,16], in Section 2.2.2 and Section 2.3.1, for further details on the mechanisms implied. 

Several families of materials are adapted for (bio)degradability purposes. Surprizingly, silicon offers such properties when used as thin films. For example, Kang et al. [17] focused on Si-based nanomaterials (e.g., silicon nanomembranes) which can degrade by hydrolysis in biofluids, and are therefore a new class of bioresorbable electronics, which are already very competitive because they benefit from the actual Si technology, conversely to alternatives such as organic semiconductors, which not only present intrinsic lower performances but need completely renewed industrial equipment to be implemented in commercial devices. However, degradable organic materials are more developed, particularly polymers. For example, Cao and Uhrich published, in 2018, a review on biodegradable polymers, natural or synthetic, for electronic applications [18]. Feig et al. [19] also dealt with polymers, with a focus on materials which combine biodegradability with long-range conjugation, a feature which has been relatively unexplored. The authors particularly stressed the fact that, besides degradability, these materials must present at least the same (good) performances as actual materials to be truly attractive, which is a significant barrier to their spreading. Methods to make better-performing polymers were reviewed by Liu et al. [20], showing that polymers can be advantageously blended with nanocomposites to gain electric conductivity or isolating properties. For example, nanofillers can be incorporated into polymers to add a function while simultaneously maintaining effective biodegradability. X. Li et al. also published, in 2020, a review focused on molecular engineering of biopolymers for next-generation flexible and wearable bioelectronics [21]. Other natural materials are obviously pertinent. Baumgartner et al. published, in 2018, a book chapter dealing with green materials for electronics [22], focusing on paper, silk, or even more original materials such as Aloe Vera derivatives. In the same spirit, Le Borgne et al. [23] published a work where all materials were everyday life ones, from carbon soot to egg white, deposited using a family inkjet printer on basic polyethylene terephthalate slides and paper sheets. As a proof-of-concept, they fabricated an RC filter which demonstrated sufficiently good performances to make it suitable for HF reception applications. Even more excitingly, we will see at the end of Section 2.4.4 that active sensors can be entirely made from edible materials, without any use of synthetic products.

Table 2 below summarizes the various (bio)degradable materials used today in green electronics, with their chemical structure. The reader is invited to refer to this table each time these names and acronyms are cited in the core of the review.

##### Design and Fabrication Methods

Not only can materials be improved, so can the design. As the proverb says, “The electric light bulb was not invented by improving the candle”. For example, Fu et al., in 2016 [24], focused particularly on the design of electronic circuits, which may not be simply copied and pasted from conventional Si technology, but are re-engineered, taking into consideration the particular properties and performances of transient materials. Flexibility and thinness are features that are widely explored: two properties which also fit with additive technologies such as printing, which are often better adapted to the fabrication of electronic devices using solution-processable organic materials than the traditional subtractive microtechnologies used today in the Si industry. See, for example, Tan et al. [25].

For a very recent landscape of bioresorbable and biodegradable materials and devices, the reader can also refer to a review published in mid-2020 by W. Li et al. about green electronics [26], and two recent book chapters from Kuzma et al. [27] and Cheng [28].

### 2.2. Materials for Substrates, Supporting Matrices and Packaging

Before citing active electronic materials, we will review the passive ones, i.e., the substrates onto which circuits are lithographied or printed, matrices with which catalytic materials are blended, or packages used to protect electronic devices from shocks, electrical contact or humidity.

#### 2.2.1. Inorganic Materials

Among all possible materials, carbon is probably the first to be cited. Indeed, carbon materials are renewable by essence, being obtained through green processes from biomass, e.g., pyrolysis or hydrothermal carbonization [29,30], and can degrade or return to nature without further treatment, except for finely divided nanomaterials. Macroporous carbonaceous materials [31], carbon fibers [32], carbon dots [33,34], carbon nanotubes or nanohorns, etc., can be synthesized from green processes then disposed of without the need for particular depollution treatments. Carbon materials are mainly applied as a substrate or supporting matrix, in the form of simple carbon black, carbon felt or carbon fiber, but also as supporting nanoparticles such as carbon nanotubes, graphene or reduced graphene oxide sheets, etc. In the particular case of gas sensing, carbon materials can compete with more well-established inorganic materials. For example, Kim and Lee (2016) [35] and Han et al. published two reviews on the enhancement of gas sensors performances using carbon nanomaterials [36]. It appears that such supporting materials are effective due to their high surface-to-volume ratio. For the same reason, carbon materials were also exemplified as substrates or matrixes in various other applications, e.g., for capacitive or resistive humidity sensors [37], pressure and strain sensors [38], electrochemical sensors and biosensors in general [39]. Due to the processability of carbon matrices, they can be easily implemented in flexible sensors [40]. Examples will be detailed in Section 2.4.5.

As non-conventional substrates or matrices, magnetic materials are also often used. Indeed, metal oxide magnetic materials present two interesting features: except for ceramics, they are biodegradable or can be easily recollected then recycled after use with the help of an external magnetic field. Many examples were reported in the recent literature of using magnetic particles in sensors [41,42,43,44], all providing at least collection and reuse (reuse of the sensor several times) if not recyclability. It should be noted that such magnetic particles, as for carbonaceous materials, can be prepared from renewable or recycled materials.

Metals can also be used as substrates, providing that they are used as thin foils. For example, Kang et al. [45] studied transient materials such as Mo, Fe, W, or Zn as biodegradable substrates able to dissolve in water, which were demonstrated to be non-toxic and pertinent for transient electronics.

#### 2.2.2. Organic and Hybrid Materials

The most often-encountered organic materials used as substrates are biosourced or biodegradable polymers, textiles, silk, paper and cellulose-derived materials, the latter being the most frequently biosourced and biodegradable substrate used for electronics, often conjugated with printing techniques such as inkjet printing or screen printing. Obviously, the use of paper implies that the whole fabrication needs only low-temperature processes, including the curing of conductive tracks, semiconductors, or dielectrics, if any. For example, Martins et al. [46] described, in 2013, their results obtained by ink-jetting CMOS (Complementary Metal Oxide Semiconductor) circuits on paper (Figure 1). Electron and hole field-effect mobilities were shown greater than 20 and 1 cm^2^ V^–1^ s^–1^, respectively. The obtained circuits performed well at low voltages, and were therefore suitable to run low-power devices such as printed sensors and biosensors (these performances will be detailed in the last section of this review). There are many different kinds of papers, which not well-adapted to all applications. Typically, to obtain good performances, the authors demonstrated that the physico-chemical properties of the paper, e.g., the size of the fibers and their compactness, have to be adapted to the size of the electronic components which are printed or deposited onto it.

The same year, Valentini et al. [47] also published a preliminary work where they printed a MOSFET device on a paper substrate, obtaining encouraging results (i.e., typical transfer curves, as expected). The semiconductor was poly(3-hexylthiophene) (P3HT) and the gate’s dielectric material was graphene oxide (GO). In 2013, Yang et al. [48] made a NO_2_ gas sensor on paper. Silver paste was used for the conductive tracks and graphene as an active material. The authors insisted on the fact that the device was poorly sensitive to strain applied on the substrate (which is very dependent on the paper’s quality, however) but did not investigate the effect of deformation. Peng et al. [49] combined screen-printing and thermal evaporation to fabricate arrays of organic field effect transistors onto standard printer paper. The (hole) field-effect mobility was over 0.56 cm^2^ V^–1^ s^–1^ and the on/off ratio was very high, 10^9^. The cut-off frequency was 39 kHz, which made the array suitable for display applications. Jung et al. demonstrated [50] the fabrication of a gallium arsenide flexible microwave device made on cellulose nanofibril paper with a similar performance to its rigid silicon-based counterpart (Figure 2). They also clearly demonstrated the fungal biodegradation of the cellulose-based substrates (Figure 3). This work demonstrated the possibility to fabricate ecofriendly and efficient electronics which could be used in wireless-communicating sensors.

Many other examples were reported in the literature, for example, that of Kanaparthi et al. [51], which described the fabrication of temperature and infrared sensors on cellulose filter paper with a carbon nanotube-based ink as a sensing surface and graphite for the conductive tracks. The temperature sensor displayed a temperature coefficient of resistance (TCR) comparable to available commercial temperature sensors (between −3100 and −4900 ppm K^−1^) while the IR sensor shows a high responsivity of 58.5 V W^−1^. Guna et al. published, in 2016 [52], a completely biodegradable printed circuit board (PCB) from cellulose extracted from banana stems and wheat gluten, normally considered as agricultural waste. The dielectric constant of the material was varied, depending on the content, between 2 and 36, in the range of conventional dielectrics for PCB. The board was also able to dissipate heat, as its epoxy counterparts were resistant up to 100 °C and were poorly (but not insensitive) to high humidity levels. Liu et al. [53] have shown the fabrication of inorganic flexible indium oxide and silicon-based nanowire transistors on paper substrates, using low-temperature processes. The transistors operated at a low voltage (1 V), with subthreshold swing, current on/off ratio, and field-effect mobility of 74 mV/decade, 1.7 × 10^6^, and 218 cm^2^ V^−1^·s^−1^, respectively, which make them applicable in portable flexible sensors. For many other examples, in particular, those based on organic electronics deposited on paper, the reader can refer to the review published by Zschieschang and Klauk in 2019 [54]. Zhu et al. also published a review of cellulosic materials for green electronics [55], where they particularly focused on structure–properties–application relationships, showing the importance of controlling the micro- or nanostructure of the substrate material. This is true for conventional paper, but also for more elaborate wood-derived materials. For example, in a particularly rich article, Fu et al. [56] described a wood-based flexible and transparent substrate which they used to make a printed strain sensor. To make this material, they removed lignin and hemicellulose to nanostructure the material and collapse the cell walls, preserving the original alignment of the cellulose nanofibers and promoting their binding (Figure 4 and Figure 5). In the fiber direction, the Young’s modulus and tensile strength were high, 49.9 GPa and 469.9 MPa, respectively and the strain sensor’s properties were very convincing (Figure 6). Even the conductive carbon ink was made from carbonized wood.

Nanocellulose (nanopaper) is a material which quickly attracted the interest, due to its additional properties compared to traditional paper, including transparency, gas impermeability and improved mechanical properties. For example, Gaspar et al. [57] reported the use of a cotton-based nanocrystalline cellulose as substrate and gate dielectric layer to print field-effect transistors, with the channel being made of an oxide amorphous semiconductor and the gate electrode of a transparent conductive oxide. This hybrid FET presented high field-effect mobility (7 cm^2^ V^−1^·s^−1^), an on/off ratio above 10^5^ and a subthreshold swing over 2 V/decade. The authors proposed to use such a device and substrate for point-of-care sensors. For more details, the reader can refer to the recent review from Zhao et al. [58], who focused on the molecular structure and nanostructures (nanocrystals, nanofibers, nanosheets), and the processing technologies for fabricating cellulose-based flexible electronics. Miyashiro et al. also reviewed cellulose-based materials, more particularly mixed materials of cellulose and carbon nanotubes [59], for electronic applications.

Apart from wood derivatives, other type of fibers can be used as well, in the form of various textiles and fabrics, such as those reviewed by Kamarudin et al. [60], who discussed the most recent textiles used in electronics, their structure, novel processing technologies and the remaining challenges. Among the emerging and original substrates, silk is one of the most studied. As soon as 2009, Kim et al. [61] reported strategies for integrating thin-film silicon devices onto silk substrates. They studied mechanical resistance, water dissolution and biocompatibility, and suggested that silk offers promising opportunities, in particular for implanted biomedical sensors. As a remarkable example of the use of silk for implantable and resorbable electronics, the reader can also refer to the work of Hwang et al. in 2012 [62] (Figure 7). In their device, they used doped single-crystalline silicon membranes (0.3 μm thick) obtained from silicon-on-insulator (SOI) wafers. The release of silicon from the SOI was realized by wet etching with HF, then the membranes were transferred to a spin-cast film of silk on a Si wafer. Silk was obtained directly from silkworm cocoons. After a dedicated preparation, the silk solution in water was cast onto Si substrates to give 20-μm-thick films. Interlayer dielectrics or metallic interconnects were deposited by electron-beam evaporation through a mask on this silk film. The packaging (encapsulation) layers (under and upper layers) were also made with 100-μm silk fibroin films.

Biodegradable natural or synthetic polymers are another family of transient substrates. The reader is invited to refer to Yin et al. [63], who recently reviewed the classifications and the various applications of modern biodegradable polymers, with a focus on the development of functions added to these polymers. Hwang et al., after working on silk, also worked on such synthetic polymers. In particular, they proposed the separation of the fabrication processes dedicated to the electronic components from the transfer processes to the substrate, which allow the active components to be processed at a high temperature or stringent conditions, separately, even if the final substrate is poorly resistant, which is generally the case in biodegradable ones [64]. More precisely, components and conductive tracks were processed and deposited on a resistant but temporary substrate, then transferred to the fragile definitive biodegradable one. They demonstrated this approach with poly(lactic-co-glycolic acid) (PLGA), polylactic acid (PLA), polycaprolactone (PCL) and also a naturally sourced rice paper, which also used a polyimide layer, but one obtained from the diluted polymer (D-PI). The authors demonstrated the process by making a hydration sensor and also showed that the approach allowed transfer even on nonplanar surfaces. The resulting electronic functions were of excellent quality (Figure 8) and degradability was excellent as well (Figure 9).

Another very good example of what could be done with dissolvable electronics was also given by the same group [45] with silicate spin-on-glass (SOG) materials (Figure 10, Figure 11 and Figure 12).

Maccagnani et al. published, in 2019 [65], another approach to preparing flexible and transparent metallic films by sputtering thin gold layers on a sodium alginate free-standing substrate. The resulting sheets demonstrated excellent resistance to mechanical stress and were stable in ambient atmosphere over several months. Disassembly of such bilayer films was easy by dissolving the alginate layer in water and collecting the supernatant gold film. Recently, Harnois et al. [66] also investigated the transfer of inorganic electronic materials (conductors, semiconductors, dielectrics) onto degradable substrates. In particular, silicon-based elements were transferred from PI to PVA films. They demonstrated that the transfer stage does not affect the electrical characteristics of the devices. After dissolution of PVA in water, the active materials can be recollected and recycled, as in Maccagnani et al. Less conventional materials were also investigated. For example, Rullyani et al. reported, in 2018, a flexible biodegradable and biosourced polymer as a substrate for electronics [67], namely polypropylene carbonate (PPC), obtained from the up-cycling of CO_2_. They utilized casted thin films of this material as a dielectric and substrate for OFETs. Its dielectric constant was reported as low, however (around 3), leading to a high operating voltage of 60 V. With pentacene as a semiconductor, electron and hole’s field-effect mobility were estimated at 0.14 and 0.026 cm^2^ V^−1^·s^−1^, respectively, and the on/off ratio above 10^5^. The fabricated PPC sheets had acceptable mechanical properties and, most importantly, biodegraded rapidly in a medium containing a lipase enzyme from *Rhizhopus oryzae*.

The successful fabrication of sensors of a comparable performance to Si-based ones, on natural fiber substrates such as paper, silk or other textiles, but also on degradable or dissolvable polymer thin films substrates, demonstrates that it is possible to fabricate high-performance electronics using eco-friendly materials, often at low cost, without or with minimal need for conventional clean-room technologies.

Degradable organic materials can also be used as matrices, e.g., for the encapsulation of bioactive components in biosensors. For that purpose, hydrogels are indicated. The reader can refer to the review from Mondal et al. [68], which details the recent developments on polymer hydrogels for various applications, including biosensing. Stimuli-responsive hydrogels were investigated, as well as conducting polymer hydrogels, polymer hybrid gels containing carbon nanomaterials, peptide gels, etc. As an example, one may cite the work of Hwang et al. [69], who described a dopamine sensor made of a fully recyclable agarose hydrogel serving as a matrix for the active sensing part made of polypyrrole, carbon nanotubes and DNA aptamers. For recycling, the hydrogel was dissolved in hot water, the resulting suspension centrifuged, and the precipitate re-introduced in a new agarose hydrogel for a new cycle. More elaborated is the work of Cunha et al. [70] who proposed a reusable eco-friendly hydrogel electrolyte based on cellulose, applicable to flexible electrochemical devices. The gel was made of microcrystalline cellulose in aqueous LiOH/urea. The process produced free-standing flexible electrolyte films with high specific capacitances (4–5 µF·cm^−2^). As a demonstration, IGZO (Indium Gallium Zinc oxide)-based, electrolyte-gated transistors (EGTs) were made with such cellulose-based hydrogel electrolytes (Figure 13). Low working voltages (<2 V) were obtained, with an on/off ratio above 10^6^ and a subthreshold swing of 0.2 V·dec^−1^. The electron field-effect mobility was ca. 26 cm^2^·V^−1^·s^−1^ and the cut-off frequency was ca. 100 Hz (Figure 14).

Wang et al. [71] very recently published a promising method to fabricate biodegradable and recyclable conducting films from silver nanowires (AgNWs) embedded by a transfer method into a blend of poly(L-lactic acid) (PLLA) and poly(D-lactic acid) (PDLA). Such a transparent composite film presented high electronic conductivity, good heat resistance and good mechanical strength and flexibility, and underwent facile hydrolytic degradation (Figure 15).

Cellulose hydrogels are also very much utilized for making biodegradable electronic devices and sensors. The reader can refer to the reviews from Teeri et al. [72] or Kabir et al., 2018 [73], which detail the physico-chemical aspects of these hydrogels based on cellulose, chitin, or chitosan. These polymers are responsive to pH, temperature, or even chemical species, and are biocompatible, biodegradable and abundant. These properties can serve in different kind of devices such as humidity sensors, strain or pressure sensors, or even chemical sensors. As an example, Pinming et al., in 2016, proposed a resistive humidity sensor using carboxymethylcellulose cross-linked with epichlorohydrin as the sensitive membrane [74]. Its electrical resistance varied satisfactorily for high relative humidity (RH) levels between 50 and 95%. Similarly, Wang et al. described a cellulose ionic film as a humidity sensor [75] which showed fast and reversible response to RH in the range 11–97%. Its response and recovery times were 6 and 11 s, respectively, with a small hysteresis of less than 1% (Figure 16). To produce the cellulose ionic film, pristine cellulose was dispersed in benzyltrimethylammonium hydroxide, and the result was cast onto a glass plate and left to dry, then washed with water and immersed in 4 wt. % KOH to obtain the cellulose/KOH composite hydrogel film. The authors demonstrated non-contact fingertip moisture detection and breathing rate detection.

Cellulose hydrogels were also often reported in pressure or strain sensors. Jing et al. [76] reported a stretchable and self-healing polyvinyl alcohol/cellulose hydrogel in a pressure and strain sensor, for motion detection. This hydrogel contained three kinds of bond which contributed to its mechanical resistance (Young modulus of 11.2 kPa; elongation rate of 1900%) and self-healing ability (within 15 s): borate bonds, metal–carboxylate coordination bonds, and hydrogen bonds. The authors showed that a thin film of this hydrogel was sensitive to the pressure exerted by a single drop of water and was capable of monitoring motions such as finger or knee movements or breathing. This kind of material is obviously very promising as smart, biodegradable, electronic skin (Figure 17 and Figure 18).

For the same application, Tong et al. [77] proposed a conductive strain sensor prepared by copolymerization of allyl cellulose and acrylic acid. The obtained hydrogel was highly stretchable (strain at break of 142%), and able to transduce strain into an electrical resistance change in a wide linear range up to 100% elongation, for over 1000 cycles. The authors applied this material to the detection of body movements. Zhou et al. also proposed a cellulose hydrogel as pressure sensor [78]. With a high content of Ca^2+^ ions, it was rigid (compressive strength of 2.2 MPa), while a high content of Zn^2+^ ions gave a fluid state instead. It was shown that the Ca^2+^-containing gel could monitor slight bending and pressure changes, e.g., of fingers. Tong et al. [79] also described a stretchable and compressible cellulosic hydrogel (strain at break of 126%, compression strain of 80%) obtained by free radical polymerization of allyl cellulose. Its ionic conductivity change (ca. 0.16 mS cm^−1^ at rest) could serve to monitor strain. Pang et al. [80] proposed a skin-inspired cellulose conductive hydrogel with a self-healing performance (healing efficiency of 96.3% within 60 min) with a broad strain window (0–2000%), which could monitor both small and large motions. It also presented a good thermal sensitivity with a 10-fold increase in its conductivity upon a temperature increase of 50 °C. Huang et al. [81] described a cellulose piezoresistive hydrogel for strain and pressure sensing. It was obtained from a mixture of PVA, sodium alginate (SA), bacterial cellulose (BC), modified carbon nanotube and carbon black (MCC). The authors demonstrated that their sensor was able to distinguish between strain and pressure, and the measures were very repeatable. It was used for monitoring limb movements, walking or grasping weights (Figure 19 and Figure 20).

Cellulose hydrogels may also be designed for chemical sensing. For example, Kim et al. [82] described a cellulose/β-cyclodextrin nanofiber patch as a wearable skin glucose sensor where glucose oxidase enzymes were entrapped into a cellulose/β-cyclodextrin electrospun membrane. Detection was based on reverse iontophoresis of interstitial fluid through the skin, to monitor glucose levels between 1 μM and 1 mM, with a sensitivity of about 5 μA·mM^−1^.

Conventional packaging (encapsulations) of electronic devices could also be advantageously replaced by biodegradable materials, such as cellulosic ones. Nie et al. [83] reviewed the use of cellulose nanofibrils-based thermally conductive composites for packaging electronic elements. They focused on the thermal conductivity of the polymer composites (i.e., its ability to dissipate heat), and the type and loading of the filler. As another example, Ma et al. recently proposed a nanofibril cellulose/MgO/reduced graphene oxide (rGO) composite presenting both thermal conductivity and electrical insulation [84]. Graphene and rGO are known to present high thermal conductivity, and are therefore widely used in the field of thermal management. However, they are also good electrical conductors, which is an obvious obstacle when applying them in electronic packaging where electrical insulation is needed. Here, the authors proposed to use rGO along with MgO particles as a thermally conducting filler, entrapped into nanofibril cellulose films via mechanical compression. MgO was used for two purposes: (i) it participates in reducing the thermal resistance; (ii) it also breaks the conductive pathways between rGO sheets so as to keep a low electrical conductivity. Due to the fabrication protocol, the films exhibited high anisotropy. The in-plane and cross-plane thermal conductivities were 7.45 and 0.32 W·m^−1^·K^−1^, respectively, with a filler content of 20 wt.%, and an electrical resistivity above 1 kΩ·m. Exemplified on a light-emitting-diode, the composite film was shown to efficiently dissipate heat.

### 2.3. Active Materials (Conductors, Semiconductors, Dielectrics)

#### 2.3.1. Inorganic Materials

The reader is invited to refer to existing reviews on transient active electronic materials, such as that of Fu et al., where they partly focused on dissolvable metals [17], that of Cheng on dissolvable inorganic electronics, including conductors and semiconductors [28,85], of Li et al. in 2018 [86], of Seo et al. on hydrolysis of a nanoscale silicon surface [87] or that of Liu et al., which focuses on dissolvable inorganic thin films [88]. The shape and surface-to-volume ratio of the materials obviously influence their ability to be dissolved. This is why nanostructures, such as nanowires, are advantageous for recyclability. For example, Yang et al. [89] proposed, in 2011, a method to prepare printable and recyclable AgNWs which could be applied for interconnections in flexible electronics. They were able to form conductive tracks with conductivities up to 5 × 10^6^ S·m^−1^, which makes them competitive for real application. If not nanostructured, the materials are expected to be thin. The case of dissolvable metallic conductive tracks has been already discussed above, for example, with the work of Rogers and co-workers [46,63,65]. In particular, Hwang et al. [63] made a very nice practical demonstration of a transient device where the conducting tracks were made of Mg and the dielectric part of MgO; only the transistor channels were made of silicon (Figure 1). On their platform, all components (inductors, capacitors, resistors, diodes, transistors, conductive tracks, etc.) dissolved when immersed in water. Yin et al. [90] published a detailed work dealing with the dissolution of thin films of materials, such as Mg, AZ31B Mg alloy, Zn, Fe, W, and Mo, in water and also in simulated body fluids, to assess the potential use of these metals in transient, possibly implantable, electronics. One of their conclusions is that the dissolution rates of thin films are very dependent on their thickness, which helps to anticipate the lifetimes of the circuits beforehand. Kang et al. reported, in 2015, a comprehensive study dedicated to the dissolution mechanisms of polycrystalline silicon, amorphous silicon, silicon–germanium, and germanium in aqueous solutions at various pHs, in view of making dissolvable electronic devices (Figure 21), based on the dissolution reactions of Si and Ge in water: Si + 4 H_2_O → Si(OH)_4(aq)_ + 2 H_2_ and Ge + O_2(aq)_ + H_2_O → H_2_GeO_3(aq)_. Typically, in similar conditions, Si dissolves faster than SiGe, which dissolves faster than Ge. They also studied the toxicity of the end products of dissolution [91].

Following a totally different approach and to end this section, one may cite the work of Hu et al. [92], who described a material which is able to regenerate by itself, without the need for a mechanical recycling step. They developed self-cleaning electrodes made of carbon-doped TiO_2_ nanotube arrays (C-doped TiO_2_-NTAs) which, beyond their properties for the selective electrooxidation of (for example) ascorbic acid, can be easily photocatalytically refreshed to be immediately reused. Because of the high photocatalytic activity of the C-doped TiO_2_-NTAs electrode, the electrode surface can be readily regenerated by ultraviolet or visible light irradiation. This photoassisted regenerating technique did not damage the electrode microstructure. The authors published exactly the same work later in 2016 [93]. The idea is interesting for chemical sensors, but it is obviously not recyclable in its genuine sense.

#### 2.3.2. Organic Materials

Recyclable biosourced organic semiconductors are the first to be cited. Among them, melanin is one of the first reported natural and biodegradable semiconductors, with the charge transport properties and mechanisms offered by this molecule having been described for many years; however, these mechanisms have been revised recently. Bettinger et al. [94] proposed, in 2009, the use of natural pigments such as melanin as organic conductors or semiconductors for application in electronics, more particularly here for tissue engineering applications. Melanin thin films showed an electrical conductivity of 7 × 10^−5^ S·cm^−1^ in the hydrated state, were biocompatible and resorbed after 8 weeks, which makes them a promising biodegradable semiconducting biomaterial. However, Mostert et al. published a critical work in 2012 [95] which questioned the conductivity model in melanin and related materials. Wünsche et al. [96] did the same for eumelanin and showed that the conducting property of hydrated eumelanin films is dominated by ionic conduction (10^−4^–10^−3^ S·cm^−1^), and attributable to electrochemical processes rather than electronic conduction. Far from disqualifying eumelanin and related molecules for electronic purposes, however, this strengthens their potential for implantable and bioresorbable devices. As another example, Di Mauro et al. [97] studied the electrochemical processes occurring at eumelanin/metal contact interfaces. They reported on the chemical and structural changes occurring at interfaces between Pd, Cu, Fe, Ni and Au metal electrodes and hydrated films of eumelanin, which helps to select the best adapted metal/semiconductor couple depending on the application. There are, to date, very few examples of devices using the semiconducting properties of (eu)melanin, however. Conversely, Indigo and its derivatives were reported as true semiconductors, as detailed in the review from Głowacki et al., published in 2012 [98]. They showed that, due to hydrogen bonding and π-stacking, these molecules form highly ordered crystalline materials, as other popular synthetic semiconducting polymers do (i.e., all actual thiophene derivatives), allowing excellent electrical performances. Charge mobilities were found in the range 0.4–100 cm^2^ V^−1^·s^−1^, which make them competitive for applications in electronic devices (Figure 22). These authors have clearly related the highly ordered structure with their high (ambipolar) mobilities. In addition, dibromoindigo is air-stable and paves the way for applications in environmental conditions.

This family of indigoid materials could also inspire the synthesis of larger synthetic molecules or polymers with a similar structure, for making open air-operated biodegradable electronics. Conjugated pigments are not the only natural molecules showing semiconducting properties. Tao et al. [99] have shown that peptide supramolecular structures could provide semiconducting properties, which may serve as an alternative source for the semiconductor industry. Such assemblies can also allow the addition of electron donors and acceptors within the structure to finely tune the electronic properties and make p-n junctions, for example (Figure 23).

Another indication that such supramolecular assemblies behave as semiconductors is that FF (diphenylalanine) crystallized nanowires exhibit increased electric conductivity upon temperature increase. The superstructure geometry can affect the electrical properties. For example, long and straight tubes self-assembled by the AAKLVFF heptapeptide give the most conductive material, while shorter or curved assemblies give lower conductivities [103]. On the other side, if aromatic moieties are added as side-chain groups, the conductivity is improved [104]. It was also shown that these materials present an ionic (protons) conductivity, which could be used for humidity sensors, for example. Such peptide-based supramolecular semiconductors are still in their infancy, however, and our understanding of the mechanisms implied has to be significantly strengthened before these materials more widely utilized.

Supramolecular structures are not only obtained from engineered peptides assemblies, but also from natural proteins. In this case, however, it is their isolating properties which are exploited. For example, Wang et al. published, in 2011 [105], a work on silk fibroin, which was shown to be an excellent gate dielectric material for OFETs. Their pentacene OFET exhibited a mobility of 23.2 cm^2^·V^−1^·s^−1^ and a low operating voltage of −3 V. Using a natural material, Zhang et al. published, in 2016, an example of a rice-derived gate dielectric [106] integrated into a top gate OFET, with promising results (Figure 24).

Still using natural materials, Gaspar et al. published, in 2014, a cotton-based cellulose dielectric [58] used as both a substrate and gate dielectric in an oxide amorphous semiconductor FET. With an electron mobility above 7 cm^2^ V^−1^·s^−1^ and an on/off ratio above 10^5^, the device presented an excellent operational stability after two weeks in air, without any type of encapsulation. Valentini et al. also reported cellulose nanocrystals thin films as gate dielectrics [107], as well as Dai. et al. in 2018 [108], who not only focused on the chemical stability of the electrical properties of such a material when used in ambient conditions, but also demonstrate an interest in increasing the mechanical resistance of devices such as FET upon repetitive bending. More original was the work of Seck et al., who published an almond gum dielectric to be used in an OFET [109]. Almond gum is a natural and biodegradable material, which is water-soluble, which they show to present good dielectric properties. A bottom gate/bottom contact p-channel OFETs was proposed using this almond gum along with a conventional poly(3,6-di (2-thien-5-yl)-2,5-di (2-octyldodecyl)-pyrrolo [3,4-c] pyrrole-1,4-dione)thieno [3,2-b] thiophene) (DPPTTT)/PMMA blend as a semiconductor. The resulting transistors operated at a low voltage (<3 V), and field-effect mobilities were found above 0.75 cm^2^·V^−1^·s^−1^, with an on/off ratio of about 10^3^. Still using natural materials, Shin et al. demonstrated the possibility to make good dielectrics with semi-synthetic polymers based on natural tannic acid [110], which keep the biodegradability properties. As an example, they developed a naturally degradable poly(methacrylate tannic acid) presenting good dielectric properties. Placed in a pentacene OFET, a field-effect mobility of 0.2 cm^2^ V^−1^·s^−1^ was found at a (quite high) gate voltage of −20 V. Most importantly, with PVA as substrate, the device was fully decomposed within 8 days in phosphate-buffered saline (PBS). Last, but not least, dielectric materials were also obtained from DNA derivatives. For example, Singh and Sariciftci reported in 2006 the use of waste products such as salmon milt and egg mixtures to make OFET gate dielectrics [111]. Their OFET exhibited current-voltage characteristics comparable with OFETs using conventional organic-based dielectrics: an operating potential of 10 V and an on/off ratio of about 10^3^. Stadler et al. published on the same topic in 2007 [112], describing DNA biopolymers as a gate dielectric in n-type and p-type OFETs working at a low voltage; they evidenced a high level of hysteresis, however, which was suppressed only by adding a conventional oxide dielectric. Yumasak et al. also reported the same drawback with DNA-based biopolymers as dielectric; they show that this hysteresis is reduced by crosslinking DNA [113].

In a totally different perspective, synthetic polymers and supramolecular assemblies can be engineered so as to provide biodegradation or, simply, chemical degradation properties, e.g., by adding reversible cross-linking bonds within the structure. Following this strategy, recyclable dielectrics were also reported, for example by Luo et al., who published, in 2019, such a dielectric [114] in the form of a reversibly cross-linked composite from a maleimide-functionalized polyhedral oligomeric silsesquioxane (POSS) and an aromatic polyamide functionalized with furan groups. The reversible Diels–Alder reaction between maleimide and furan groups allowed for the cross-linking of the two polymers to reach the best dielectric properties while in function, and subsequent efficient recycling after use (Figure 25).

### 2.4. Devices

The previous sections focused on materials. This section will focus on the sensing devices obtained using these materials. Before focusing on sensors as a whole, we will have a look at elemental electronic components such as resistors, capacitors, diodes and transistors. The reader is invited to refer to recently published reviews which focus more particularly on these components, for example that of Li et al. [90], published in 2018, or that of Kang et al. [115], published in 2020. We will also review biofuel cells, expected to power implanted or external sensors, which also require (bio)degradability.

#### 2.4.1. Resistors and Capacitors

The simplest device are resistors; they can easily be made of carbon (e.g., the conventional carbon composition resistors (CCR) from the electronic industry) so that most of the existing ones could be readily degradable and recyclable if their packaging is itself degradable. However, original fabrication methods of recyclable resistors can be found in the recent literature. For example, Kumar et al. [116] proposed the printing of resistors on paper from a graphene-based ink, deposited using the bar-coating technique, with precise adjustment of the resistance value using a laser which precisely adjusts the width of the printed pattern. This kind of resistor was able to handle about 7 W at room temperature for up to 200 V.

Conversely to resistors, there are a few examples of degradable or biodegradable capacitors, with most of the effort placed on targeting electrolytic supercapacitors (EDLCs), and, more particularly, the electrolyte material. The reader can refer to the book chapter from Okonkwo et al., who reviewed biopolymers and composites for energy storage, with a focus on capacitors [117]. Lim et al. reported, in 2014, the fabrication of an electric double-layer capacitor made of activated carbon (for the electrodes) and a biodegradable water-based polymer for the electrolyte [118] which was prepared from PVA, lithium perchlorate (LiClO_4_) and antimony trioxide (Sb_2_O_3_). Cyclic voltammetry showed a conventional rectangular shape, without any redox behavior, typical of a true EDLC. The cyclability was weak, however, with 90% retention after only 200 cycles. Lee et al. went further and reported a fully biodegradable supercapacitor using water-soluble metal electrodes (tungsten, iron or molybdenum) and agarose gel as a polymer electrolyte, deposited onto a biodegradable PLGA substrate [119]. The authors investigated the dissolution kinetics and mechanisms for each individual component of the capacitor and concluded that a fine control of the composition (more particularly, molecular weights and thicknesses) could finely tune the lifetime of the device (Figure 26).

The dissolution kinetics of their thin metal films were studied, to better understand the process. This metals dissolution follows conventional hydrolysis: W + 4H_2_O → WO_4_^2−^ + 8H^+^, Mo + 4H_2_O → MoO_4_^2−^ + 8H^+^ + 6e^−^ and 4Fe + 3O_2_ + 10 H_2_O → 4Fe(OH)_4_^−^ + 4H^+^, leading to resorption of the electrodes after a few hours in water, which progressively disappear within several weeks (Figure 27).

Yang et al. proposed a fully degradable gel supercapacitor based on Zn nanosheets–Ti_3_C_2_ MXene [120]. The capacitor retains more than 80% of its capacitance after 1000 cycles and is stable at room temperature, while it is totally degraded within 7 days under heating at 85 °C (Figure 28).

Focusing on the electrolyte rather than on electrodes, Hamsan et al. proposed a potato starch-methyl cellulose blend-based polymer electrolyte to be applied in a double-layer capacitor [121]. The electrolyte was ammonium nitrate, and glycerol was used as plasticizer. The performances were good, with a potential window of 1.9 V, a capacitance of 31 F g^−1^ and a cyclability reaching 1000 cycles. Still from starch, Kasturi et al. described, in 2019, the preparation of starch-based carbon electrodes and biopolymer electrolyte for a solid-state EDLC [122]. They obtained the porous carbon from carbonized starch, while the biodegradable biopolymer electrolyte was prepared from starch as well. Their supercapacitor showed a high specific capacitance of 240 F g^−1^ for more than 2000 cycles, along with a specific energy of 17 Wh kg^−1^ and a specific power of 3823 W kg^−1^. The device totally degraded within weeks when buried in soil. More conventional was the work of Nowacki et al., who described the synthesis and characterization of membranes of chitosan [123] crosslinked with glutaraldehyde and used in EDLCs with lithium acetate as an electrolyte. Capacitances reached 106 F g^−1^ after 10,000 charge/discharge cycles. Avila-Niño et al. published, in 2020, a work where they studied the capacitive properties of gel-type natural abundant polymers such as gelatin and agar in deionized water, mounted between ITO electrodes. They obtained capacitances in the mF.g^−1^ range and good cyclability, with the device retaining 100% of its initial capacity after 1000 cycles [124]. Menzel et al. [125] also described a capacitor based on an agar gel and carbon electrodes. As for Avila- Niño et al., recyclability was not detailed, even if all constituents of the device were intrinsically degradable.

Beyond degradability is bioresorption, which is more challenging to achieve without the production of toxic side-products. Li et al. proposed a bioresorbable capacitor for implantable biosensors [126]. On a PLA-supporting substrate, the device was made by the layer-by-layer assembly of a PLA nanopillar array, a self-assembled nanoporous ZnO layer, and a PVA/phosphate buffer solution hydrogel. The capacitor was able to work in air or in liquid environment, e.g., in an animal body. The functional time was tunable from days to weeks depending on the encapsulation layer and could be fully degraded in vivo.

When developing a portable sensing device, the performance of the power source is crucial. It is now a real challenge to develop, for the purpose of application in degradable or even bioresorbable sensors, power sources with good performances and degradability/bioresorption abilities at the same time; it is a promising theme of research in the coming years.

#### 2.4.2. Transistors, Oscillators, Logic Gates

(Bio)degradable transistors or related circuits have been developing in recent years, and there are several good examples reported in the literature. For example, Guo et al. described, in 2015, a biodegradable transistor [127] from a free-standing sodium alginate membrane acting as both a substrate and dielectric, and Al:ZnO thin films as source and drain electrodes. These transistors operated at low voltage (ca. 1 V) and were completely dissolved within 1h if placed in deionized water. Rullyani et al. [128] described organic thin-film transistors by using chitosan as substrate and polyvinylpyrrolidone and natural rubber as dielectrics. With PTCDI-C8 as semiconductor, the charge mobility was about 0.03 cm^2^·V^−1^·s^−1^, with a threshold voltage of 0.8 V and an on/off ratio of 100. This transistor was used as a DNA sensor. The drawback of this approach was that the semiconducting material, PTCDI-C8, was not degradable. This was also the case for the work of Jo et al. who reported a biodegradable organic transistor using a solid-state biodegradable electrolyte [129], more precisely based on a levan polysaccharide and a choline-based ionic liquid. The electrolyte presented a specific capacitance of 40 µF·cm^−2^ and the transistor showed low operating voltage. The transistor was applied to measure the heartbeat of a rat. Yang et al. developed a double-gate InZnO synaptic transistor with wheat flour solution as an electrolyte [130]. This ion/electron-coupled transistor exhibited excellent electrical performance at operating voltages below 2 V and worked as a synaptic transistor. However, the authors did not investigate the toxicity of the dissolution products from InZnO. Martins et al. [46] went further and described low-temperature-processed p- and n- transistors based on oxide thin-films and forming CMOS inverters, deposited onto recyclable paper (Figure 29). The p-type semiconductor is SnO_2_ and the n-type is GaInZnO. In this work, recyclability was only obtained by separation and collection of the semiconductors and gold contacts after decomposition of the paper substrate, which was the only biodegradable material in the device.

Other kinds of electronic elements have been proposed, for example by Pettersson et al. [131], who described a ring oscillator based on ion-modulated transistors made from a P3HT/PLLA blend and deposited on paper. The semiconductor and insulator were mixed together in a solution and spin-casted onto a paper substrate. Due to the different surface energies, P3HT formed a thin layer on top of the PLLA one, itself in direct contact with paper. This paper-based ring-oscillator operated at 5 Hz. The authors did not directly study recyclability, but the three-layered architecture was very well adapted to that, due to its easy separation and collection of every material after disposal.

As shown here, electronics’ active components such as transistors are most often developed on the basis of partial degradability only. If substrates or even dielectrics are easily derived from natural materials and maintain their biodegradability properties, a huge effort has to be made, for semiconductors in particular, who remain poorly degradable.

#### 2.4.3. Antennas

For data transmission, recyclable antennas were reported. For example, Kim et al. described an antenna with single-ring resonators inkjet-printed on photo paper using a conductive Ag ink. The structure could separate each constituent of the antenna for easy recycling [132]. Better still, Akbari et al. proposed truly degradable graphene antenna deposited on cardboard [133] as a promising low-cost, recyclable, and flexible substrate for wireless electronics. The sheet resistance of their graphene antenna was 1.9 Ω/sq. A dipole antenna with a length of 143 mm achieved an efficiency of 40% at 889 MHz and a read range of more than 5 m at 950 MHz. In the same spirit, Kanaparthi et al. [134] described a flexible 2.4 GHz antenna on paper, with conductive tracks in aluminum and cellulose filter paper being used as a dielectric substrate. The antenna showed a stable performance after hundreds of bending cycles. The substrate can biodegrade while aluminum is recycled. In a totally original and different approach, Teng et al. demonstrated liquid metal-based transient circuits applicable for recyclable electronics [135]. They combine room-temperature liquid metals for connections with water-soluble PVA for packaging. Such circuits do not shown degradability at all but, instead, a high recycling efficiency of up to 96% of the metal used due to their liquid state. The authors made an effective transient antenna using this technique.

#### 2.4.4. Biofuel Cells

Sensors, either external or implanted in the human body, have to be powered. Some dedicated degradable batteries and supercapacitors have been cited above, but the difficulties have been underlined. Instead, and more particularly for implanted sensors, the most promising power supply is probably the biofuel cell; their biodegradability, or even bioresorption, which are very challenging, are reviewed below. The microbial fuel cells or other biofuel cells which are not intended to be coupled with wearable non-invasive sensors or to be impanted in the human body are excluded.

It should also be considered that the frontier between some amperometric biosensors and biofuel cells is blurred. Indeed, both use catalysts (most often, enzymes) to specifically oxidize a combustible (typically glucose or sucrose) and reduce the oxidizer (typically molecular oxygen). In a biosensor, the catalysts are put together on a single electrode; the presence of the analyte in an excess of oxidizer produces a current that is ideally propotional to the analyte concentration. In a biofuel cell, the catalysts are put on two separated electrodes; the presence of both combustible and oxidizer produces a voltage difference between the two electrodes, and eventually a current when the circuit is closed. To obtain nominal power from a biofuel cell, both a combustible and an oxidizer must be present in optimal concentrations. However, for a given voltage, if the concentration of one of the two reactants decreases, the measured current decreases as well. In that sense, a biofuel cell is able to play the role of sensor for the combustible concentration, providing that the oxidizer is in excess. The first example of such devices was given by E. Katz et al. in 2001 [136]; since then, a number of examples of biofuel cells serving as sensors, and even more examples of self-powered sensors, have been described.

Even if not used as self-powered biosensors, because a combustible such as glucose is always available in body fluids, biofuel cells are extremely convenient power sources for wearable or even implanted sensors. Hao et al. recently reviewed biofuel cell-powered biosensors [137]. The reader who is particularly interested in general BFCs’ functioning may refer to the very complete review by Mano and de Poulpiquet, published in 2018 [138].

The idea of making biofuel cells biodegradable is not recent. However, most of the historical works described only partly biodegradable devices, and 100% biodegradable fuel cells only appeared in the last eight years. As an example of a biodegradable biofuel cell, R. Bilewicz and coworkers reported, in 2013 and 2014, two works [139,140] where they substituted classical soluble mediators, often poorly biodegradable or even toxic, by single-walled carbon nanotubes (SWCNTs) functionalized with aryl residues, which facilitate the electron transfer between the enzyme and the electrode. The device is a very simple cell (Figure 30). The biocathode comprises enzyme (laccase, bilirubin oxidase, etc.) and SWCNTs, while the anode is made of Zn covered by a protective layer of hydrated zinc phosphate which allows ion transport and plays the role of membrane. Such a simple biofuel cell was intended to be implanted, but the practical aspects were not studied in these works.

The group of S. Minteer also envisaged the development of a biodegradable fuel cell in 2006 [141]. They built a biodegradable enzymatic device using soybean oil as the combustible. However, the fuel cell comprised a Nafion membrane and other components which were not biodegradable. In a later publication, Minteer and coworkers [142] suggested that other membrane materials could replace Nafion, such as chitosan or even nanoporous silica. At the time, no examples of such amaterials were reported in biofuel cells. Since then, however, the development of biodegradable biofuel cells has accelerated. For example, in 2014, Filip et al. proposed replacing conventional fluorinated polymers such as Nafion by PLA. Their BFC was made of a BOD biocathode combined with a fructose dehydrogenase bioanode [143] and offered performances comparable to conventional BFCs.

In view of wearable self-powered (e.g., on skin) sensors, paper-based BFCs are adapted well. Paper can even replace non-degradable materials as a membrane; therefore, such devices were developed rapidly. For example, Jenkins et al. proposed, in 2012 [144], a partly printed BFC on paper, using aldose dehydrogenase at the anode and laccase as the cathode, with the current collectors being carbon plates. However, the mediators were osmium-based complexes or other conventional inorganic mediators, which are potentially harmful. Shitanda et al. [145] reported mediator-less porous carbon inks screen-printed on paper to make a BFC with a very high-power output of 0.12 mW·cm^−2^ at 0.4 V (Figure 31), better adapted to wearable power devices. Most components of the device were degradable, and the design was well-adapted to wearable power sensors. Wu et al. [146] also published screen-printed electrodes on paper, with the substrate also acting as a membrane. With fructose dehydrogenase and bilirubin oxidase, a maximum power output of 7.9 μW was produced. No mediator was used, which makes the device fully degradable. In situations where biological fluids are not easily accessible, or carry little oxygen, air-breathing BFCs may be more pertinent. From this perspective, Ciniciato et al. [147] described air-breathing, printed paper-based biocathodes based on bilirubin oxidase, which were mediator-less. They demonstrated a stable current output (under 0.3V) over several hours. The reader particularly interested by these aspects may refer to the work of Desmet et al., published in 2015, where paper-based biofuel cells and self-powered biosensors [148] were reviewed. The reader can also refer to the work of Bandodkar and Wang, who published a very complete review dealing with wearable fuel cells, in particular, non-invasive ones intended to be placed on skin [149].

Wang and coworkers particularly developed BFCs by relying on lactate oxidation, which is the most available combustible on the skin, i.e., in sweat. In this case, the oxidizer is O_2_. For example, Jia et al. proposed a lactate/O_2_ BFC embedded in a wristband, which is able to power a watch (Figure 32) but is also intended to power a skin sensor [150].

To achieve good performances while being biocompatible, electrode materials are determined. In this domain, the work of Holzinger and coworkers is significant [151]. In particular, they developed efficient buckypaper-based BFCs for powering implantable devices. Buckypaper is a self-supported film of carbon nanotubes which looks like black paper, with excellent electronic conducting properties. A large amount of work has been published reporting the use of this material, which is probably the most successful material used in implantable BFCs. However, actual biodegradability was, as far as we know, never studied.

Very recently, Tominaga et al. published a truly biodegradable enzymatic BFC [152] made with cellulose nanofibers as a substrate, mixed with carbon MWCNTs to make electrodes. Compared to BFCs based on buckypaper, the performances were lower, with a maximum voltage and a maximum current density of 434 mV and 176 μA·cm^−2^, respectively, and a maximum power output of 27 μW·cm^−2^ (Figure 33).

Even better than the wearable and biodegradable cells, is the edible biofuel cell, described for the first time by J. Wang and coworkers [153]. The authors fabricated an alcohol sensor with only food materials, without any kind of external chemicals. Enzymes were brought by mushroom extracts for the anodic compartment (ethanol oxidase, AOx, and its natural mediator) and apple extracts for the cathodic one (polyphenol oxidase, PPO, and its natural mediator), while charcoal was used for the conductive electrode, along with vegetable oil as a binder. The device provides a voltage (V_OC_) proportional to the ethanol concentration. At saturation of ethanol, the V_OC_ is 0.24 V (Figure 34).

This example seems to be more of a proof of concept than a real demonstration of applicability; however, for micromachines and sensors that have to be ingested, typically for analysis of the digestive tract, it could certainly make the treatment of the medical act more accepted by the patient when natural materials are to be ingested rather than lithium batteries. This, and more generally bioresorbale biofuel cells, are probably promising for future developments.

#### 2.4.5. Sensors as a Whole

Let us consider now not only basic electronic elements, but sensors as a whole. Annese et al. [154] proposed, in 2014, a biodegradable pressure sensor made of gold-printed patterns on a polycaprolactone substrate. However, only the substrate was degradable. Luo et al. [155] described a more accomplished wireless pressure sensor made entirely of biodegradable materials. A Zn/Fe bilayer was used as the conducting material, and PLA and PCL as the dielectric and structural materials. Zn degrades only very slowly under normal conditions, which is why Fe was used: by galvanic coupling, it activates Zinc and increases its degradation rate. Immersed in a saline solution, the sensor remained stable and functional for more than 80h before starting to degrade. Boutry et al. [156] also published a pressure sensor made of biodegradable materials_,_ capable of detecting a weight of 5 mg. A view of the device is shown in Figure 35. It was mainly a flexible capacitor with a biodegradable elastomer of poly(glycerol sebacate) (PGS) between the top and bottom Fe-Mg electrodes. The electrodes were fabricated by casting a thin PVA adhesive layer on top of a PHB/PHV film, followed by the evaporation of Fe and Mg electrodes. It was shown that this device degrades slowly, depending on the conditions. PGS alone lost ca. 15−20% of its initial weight after a few weeks, while PHB/PHV degrades completely after 10 weeks. The authors demonstrated the performances of their sensor for continuous cardiovascular monitoring, by the acquisition of the blood pulse wave signal on carotid and femoral arteries. 

After improvements, the same authors published, in 2019, a biodegradable and flexible arterial-pulse sensor for the wireless monitoring of blood flow [157], on the same basis and with the same components as described in their 2015 publication. The device was fabricated by the lamination of soft Mg tracks, a PGS dielectric layer, a poly(octamethylene maleate anhydride citrate) (POMaC), PHB/PHV as packaging layers, and PLLA as a spacer between layers when needed. The packaging was an elastomeric POMaC layer placed in direct contact with the artery, whereas the stiffer PHB/PHV layer was in contact with the surrounding muscles, thus producing a device that is more sensitive to artery expansion than respiratory motion (all constituents’ formulas are given in Table 2). The sensor could be wrapped around arteries with a diameter smaller than 1 mm.

Hosseini et al. also reported a flexible pressure sensor [158] based on films of macrometer-sized spherules of β-glycine embedded in a chitosan polymer, with glycine forming a ferroelectric phase. Prepared by drop-casting, these biodegradable films presented a good sensitivity of about 2.8 ± 0.2 mV kPa^−1^, comparable to conventional piezoelectric materials. Using a remarkably simple approach, Kanaparthi et al. [159] reported the fabrication of a biodegradable interdigitated capacitive touch sensor just by drawing an interdigitated pattern on filter paper using a graphite pencil (Figure 36).

Zhong et al. [160] described a kind of paper-based power source which could work as a mechanical sensor. It was based on the measurement of the electrostatic charges generated by the internal movement of the device, composed of two sheets: one of PTFE/Ag/paper and the other of Ag/paper. A Kapton tape was put around the device so as to form an arch-shape with a space in-between the two sheets. The size was 2.5 × 2 cm^2^ (Figure 37). The output power of the device reached ca. 90 μW·cm^−2^. Beyond the possibility of producing power, one of the applications of such a device is as a paper-based, disposable and degradable deformation sensor. For example, the authors measured that simply turning a book page generates a current of 4.6 mA under a voltage of 2.4 V.

Wang et al. [161] reported, in 2020, a very nice work where they described a biodegradable sensor obtained from polysaccharides, which was able to monitor human breathing and ambient humidity. Au/Cr electrodes were evaporated on a degradable biocomposite of chitosan. Functionalized polysaccharides (chitin derivatives) were also spin-coated onto the device, which made the sensor easier to apply to the skin. The authors showed that it could completely decompose when placed in water (Figure 38 and Figure 39).

Biodegradable temperature sensors have been also described. One of the best examples is from Salvatore et al. [162], who reported a temperature sensor made of Mg tracks and connectors, a biodegradable elastomer (Ecoflex), silicon nitride and silicon dioxide. They studied its dissolution kinetics in water and applied it for food quality tracking (Figure 40).

It appears that relatively few (bio)degradable (bio)chemical sensors have been reported to date, while most of the sensors described are physical sensors. This can be explained by the fact that the complexity of biosensors is higher than that of their physical counterparts, comprising more components. Some examples are available, however, e.g., that of Li et al. [163], who reported, in 2020, an electrochemical flexible and biodegradable NO sensor with a low detection limit (4 nM) and a wide sensing range (0.01–100 μM), able to continuously monitor NO levels in living mammals for several days before degradation (Figure 41).

### 2.5. Fabrication Methods

Through the above sections, renewable and/or (bio)degradable materials have been reviewed, then the devices made of these materials. It was shown that not only materials, but also the design of the circuits themselves, can make devices more or less efficiently degradable. For example, the thinness of the conductive tracks, of the substrate or of the dielectrics makes them more easily and rapidly degraded. Following another approach, the layering of the components allows a more efficient separation when the objects are intended to be recycled rather than to self-degrade.

Beyond the design, the techniques used to fabricate those devices are also important. Conventional methods utilizing clean room technologies are, of course, still valid, but with the emerging technologies with a low footprint, and allowing for layer-by-layer fabrication, the deposition of thin films and use of fragile active or substrate materials (paper or other temperature-sensitive materials) may be more pertinent. For these reasons, printing technologies are gaining in importance. The reader can refer to a number of reviews focused on printing techniques in general [164,165,166,167]. There are fewer reviews which detail printing techniques for (bio)degradable devices. Kanaparthi published, in 2016 [168], a report on eco-friendly ways to fabricate flexible and degradable electronics by means of printing methods, in particular, by using degradable substrates such as paper, for all applications where size is not a constraint. Kamarudin et al. published a more recent (2020) [60] review focusing on green fabrication strategies for sensors, with the utilization of eco-friendly materials and various printing methods. Reusability is also discussed.

For the sake of biodegradability, but also for historic reasons (printing technologies were initially developed for reproducing texts on paperbooks), most printing strategies rely on paper as a substrate. For example, Peng et al. [169] described, in 2014, the screen-printing of source, drain and gate electrodes of a dinaphthothienothiophene (DNTT)/parylene C transistor matrix (Figure 42). Here, degradability concerned only the substrate and not the active materials.

More complete was the work of Bihar et al. [170], who described a fully inkjet-printed disposable glucose sensor on paper, with all components not only printed on paper, from the electronics to the biorecognition elements, but also degradable or easily recylcable. Bedük et al. [171] also published an all-inkjet-printed paper sensor made of PEDOT:PSS (not degradable), functionalized with ZnO (degradable) for the amperometric determination of hydrazine. Serpelloni et al. very recently published [172] an article where they described other methods than screen-printing or inkjet-printing, such as aerosol jet printing, to be deposited on various cellulose-based materials (chromatographic paper, photopaper, cardboard) without the intrinsic limitations of inkjet in terms of the viscosity and surface tension of the ink. They exemplified their work with multilayer capacitive sensors.

Other techniques were also described to make recyclable or degradable sensors while maintaining the precision and lateral resolution of today’s microelectronics, but, from the constatation that most often leads to degraded performances and degraded resolution, another way is to try to improve the existing photolithographic process, which also allows industrials to continue to use the existing costly equipments. Following this approach, it is not (bio)degradability which is expected, but an easier recyclability of the products. For example, Wie et al. [173] described a wafer-recyclable, environment-friendly transfer printing process which enables multiple reuses of the wafer. Controlled delamination of the active layer is enabled through a cracking phenomenon in a water environment. The layer can then be pasted onto any kind of substrate, and eventually removed after use for recycling (Figure 43).

## 3. Conclusions and Perspectives

The recycling of goods has shown limitations for electronics or for plastics. Most of them are incinerated or buried instead of being recycled, with no negligeable part being left uncollected, polluting the environment. This failure shows that collection and recycling are not sufficient and must be completed, if not replaced, by self-degradability.

Figure 44 shows the evolution of the number of publications in which the concepts of (a) green electronics, (b) degradable electronics, (c) biodegradable electronics, (d) bioresorbable electronics, (e) paper electronics, (f) hydrogel electronics, (g) biodegradable metals, (h) biodegradable sensors and (i) biodegradable pressure sensors are claimed. If the concept of green electronics was developed far before 2010, we have shown here that recyclability or degradability only relied on the use of biodegradable substrate such as paper and printing techniques, which both emerged in the beginning of the 2000s, even if the active materials deposited on these substrates were not degradable. Partly supported by the early development of biodegradable metals (for other applications), biodegradable electronics emerged after 2012, along with bioresorbable electronics, driven by the hudge demand in wearable and implantable sensors and biosensors. Biodegradable sensors (pressure sensors excluded) represent a progressively larger fraction of these biodegradable electronic devices over time, as well as biodegradable pressure sensors, which only became significant after 2016. It is a fact that the simple degradability feature, not for biomedical applications, for which the device must mandatorily disappear after a given time, but for the simple purpose of decreasing the ecological impact of modern electronics, was not a driving force until recently, even for marketing. For this reason, the corresponding sensors took more time to emerge: it is only in 2020 that the number of publications became significant. Hydrogel-based electronics were the last to emerge, mostly due to the development of degradable, biodegradable or bioresorbable biofuel cells, batteries and supercapacitors.

This review shows various aspects related to transient electronic sensors, from materials, fabrication methods, individual electronic components and architecture engineering, to whole sensors and their applications. As shown in our review, most truly transient sensors reported to date are physical sensors (temperature, humidity, pressure) and extremely few (bio)chemical sensors claim full (bio)degradability. This could be explained by the simplicity of physical sensors relative to their chemical counterparts, which generally include more functional materials. However, (bio)degradable (bio)sensors are also expected to follow the biodegradability revolution, because products on the market are increasingly regulated. Most of the technical difficulties come from the fact that several (reactive) chemical functionalities must be implemented on a chemical sensor and remain active for the lifetime of the device, while these kind of sensors, conversely to physical sensors, which can be isolated for air or water, work directly in aqueous media. It is also a fact that the best catalytic materials, which are mandatory in chemical sensors, are generally non-environmentally friendly materials, because of their intrinsic reactivity. However, significant efforts must be made quickly in that direction; one way could be to engineer new materials starting from the already known biodegradable ones (see Table 2) to which functional groups will be added, a strategy which has not yet been deeply investigated. The programmed disappearing of non-degradable plastics will probably help.

As shown, active materials such as semiconductors are still in the stone age in terms of degradability. The reason for this is probably that, to date, their electrical performances remained significantly lower than those of their conventional inorganic or organic non-degradable counterparts, so the replacement cannot be viable. However, for most sensing applications, the highest electronic performances are not needed, so these criteria should not impede the development of (bio)degradable or bioresorbable sensors.

To conclude, as underlined in this review, apart from materials, the design of the devices itself plays a role in their degradability. In that sense, sensors under the form of thin films, stickers or even tattoos, which are extremely thrifty in materials and which have been developing fast in recent years, are another part of the answer.

## Figures and Tables

**Figure 1 sensors-20-05898-f001:**
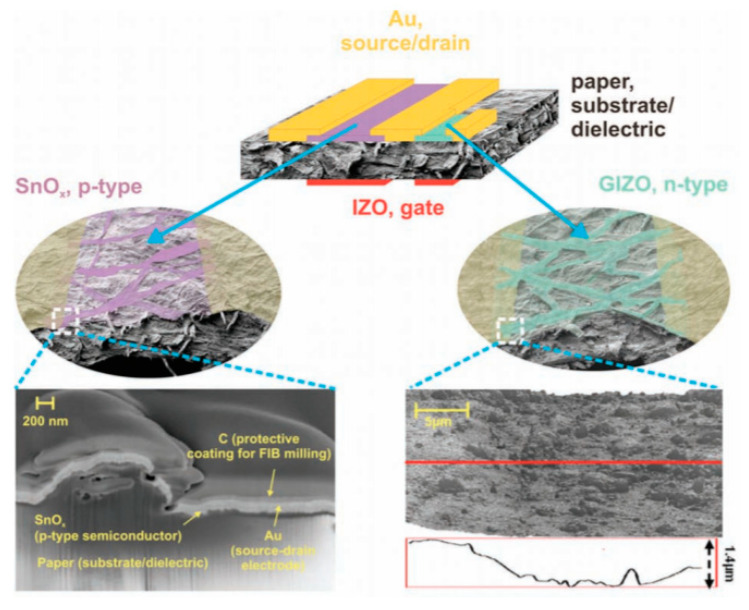
Device and paper substrate described by Martins et al. Source and drain of the two transistors composing the CMOS (Complementary Metal Oxide Semiconductor) inverter are deposited on top of a paper layer, while the indium-zinc oxide (IZO) bottom-gates are deposited on the other side. Drain and source electrodes are made of a Ni/Au film. The bottom-left figure shows a cross-sectional SEM image of a paper fiber from the p-type side, while the bottom-right figure shows an AFM (Atomic Force Microscopy) image of the section from the n-type side. Reproduced from [46] with permission. Copyright © 2012 Wiley and Sons.

**Figure 2 sensors-20-05898-f002:**
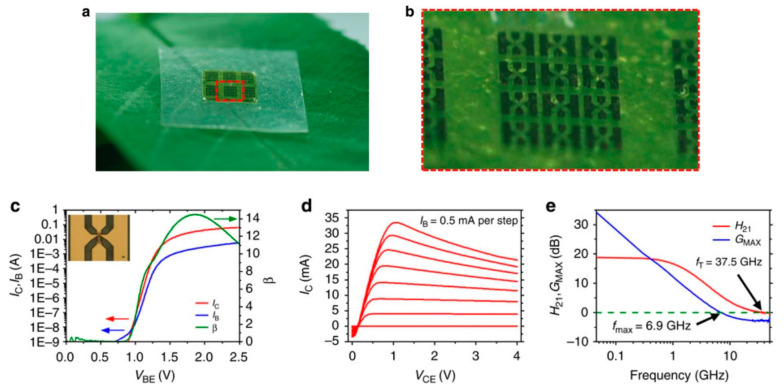
(**a**) Photograph of an array of heterojunction bipolar transistors (HBTs) on a transparent cellulose nanofibril (CNF) substrate; (**b**) Zoom on the array. (**c**) Collector and base currents showing the maximum gain of the device. (**d**) Collector current versus collector–emittor for various base current. (**e**) Gains (H for current, G for power) as a function of frequency, with a collector voltage of 2 V and a base current of 2 mA. Adapted from [50] under a Creative Commons Attribution 4.0 International License. Copyright © 2015, Springer Nature.

**Figure 3 sensors-20-05898-f003:**
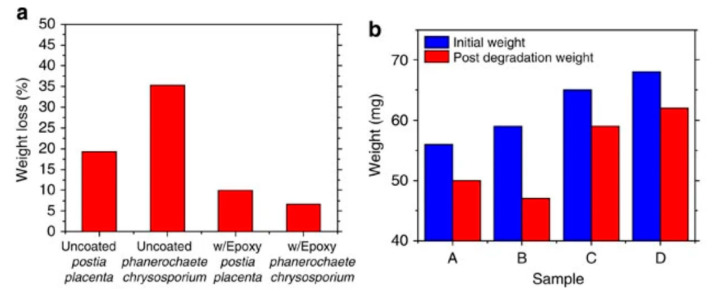
(**a**) Biodegradation tests of two types of CNF films upon addition of two kinds of fungi: left, bare CNF, and right, epoxy-protected CNF. (**b**) Weight loss upon fungal biodegradation of four different electronic devices printed on epoxy-coated CNF. (**c**) Degradation process photographed after 6 h, 10 days, 18 days and 60 days; (**d**) corresponding magnified pictures, and (**e**) magnified and tilted views of the CNF-based device after 10 days and 60 days. Adapted from [50] under a Creative Commons Attribution 4.0 International License. Copyright © 2015, Springer Nature.

**Figure 4 sensors-20-05898-f004:**
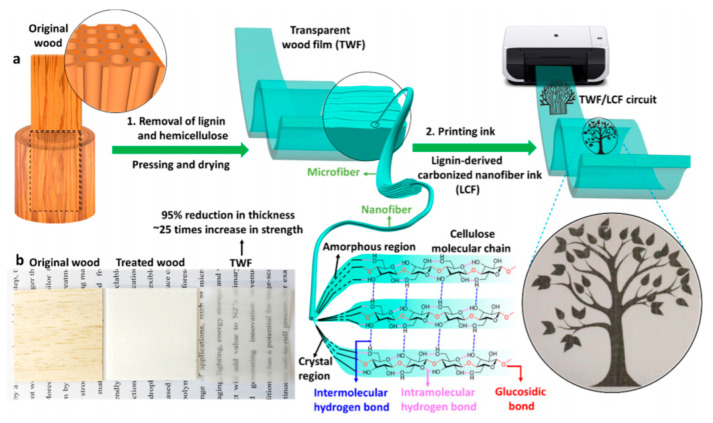
Processing of wood to make films for flexible electronics. (**a**) Lignin and half the of hemicellulose are removed from the wood, then pressed and dried under ambient conditions. Under these conditions, the chemical structure consists of cellulose microfiber bundles, nanofibrils, and cellulose chains with both crystalline and amorphous regions. Ink is also derived from wood, more precisely carbonized lignin nanofibers (LCF). (**b**) Photograph of a wood film before the removal of lignin and hemicelluose, then after removal, then after pressing. Reproduced from [56] with permission. Copyright © 2020 American Chemical Society.

**Figure 5 sensors-20-05898-f005:**
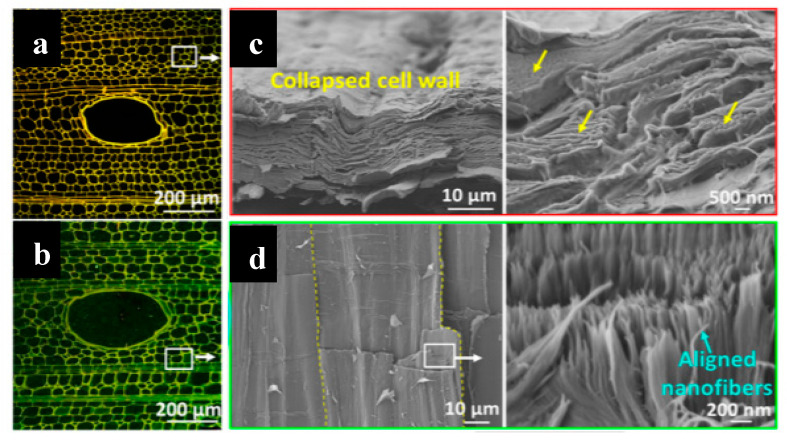
Confocal microscopy images at ×200 magnification for (**a**) untreated and (**b**) treated wood. (**c**) SEM images of a thin wood film cross-section, showing a lamellar structure of collapsed cell walls. Yellow arrows point toward lamellar cross-sections. (**d**) SEM images showing a single fiber (contoured in yellow) and cellulose microfiber bundles and nanofibrils (blue arrow). Adapted from [56] with permission. Copyright © 2020 American Chemical Society.

**Figure 6 sensors-20-05898-f006:**
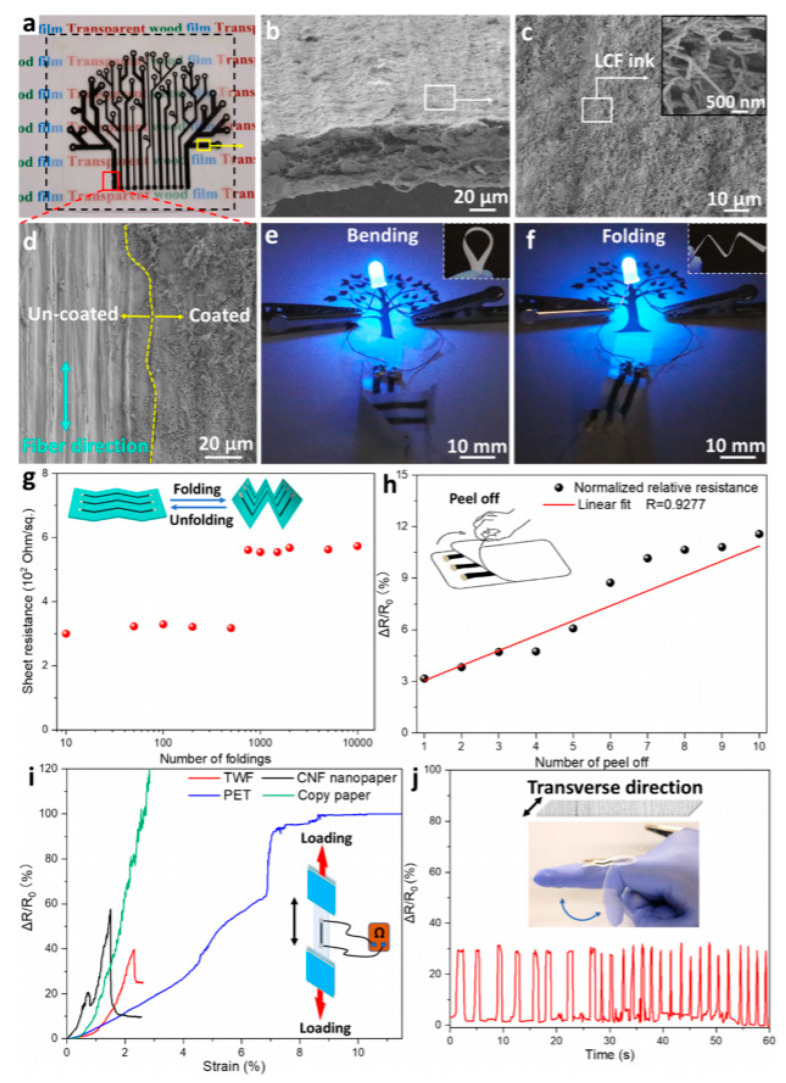
Performances of the thin wood substrate in terms of conductivity and flexibility. (**a**) Printed electronic circuit on the substrate. (**b**) and (**c**) Cross-sectional SEM view of the printed wood-derived ink. (**d**) SEM image on the edge of a printed track. LED put on for a bent (**e**) and a folded (**f**) substrate. (**g**) Sheet resistance of a printed track upon folding−unfolding cycles. (**h**) Resistance to repeated peel off. **(i)** Normalized relative resistance variation upon strain applied parallel to the fiber direction. (**j**) Normalized relative resistance variation upon repetitive 90° folding (resistance measured along the fiber direction). Reproduced from [56] with permission. Copyright © 2020 American Chemical Society.

**Figure 7 sensors-20-05898-f007:**
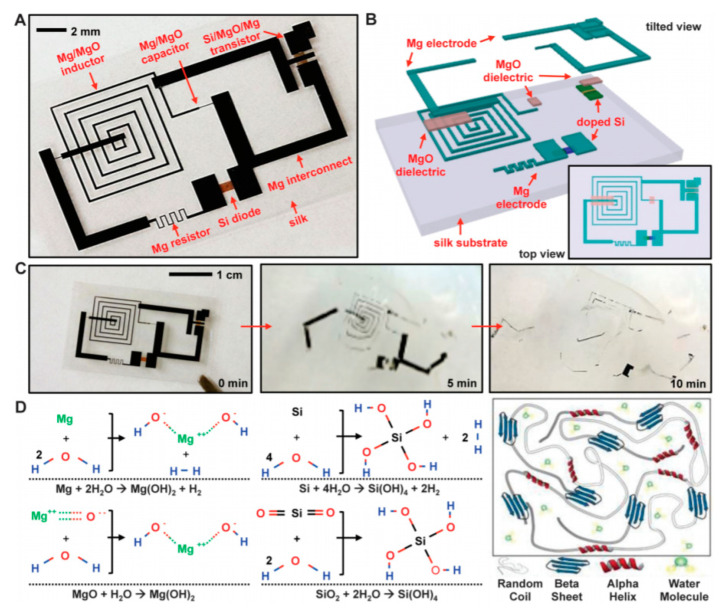
Transient electronics featuring tracks, resistor, inductor, capacitor, diode and transistor, printed on a silk substrate. (**A**) Optical image of a device. (**B**) Exploded view showing the different layers (inset: top view). (**C**) Time sequence of dissolution of the whole device in water. (**D**) Details of the chemical reactions with water for each of the constituents. Reprinted from [62] with permission from AAAS.

**Figure 8 sensors-20-05898-f008:**
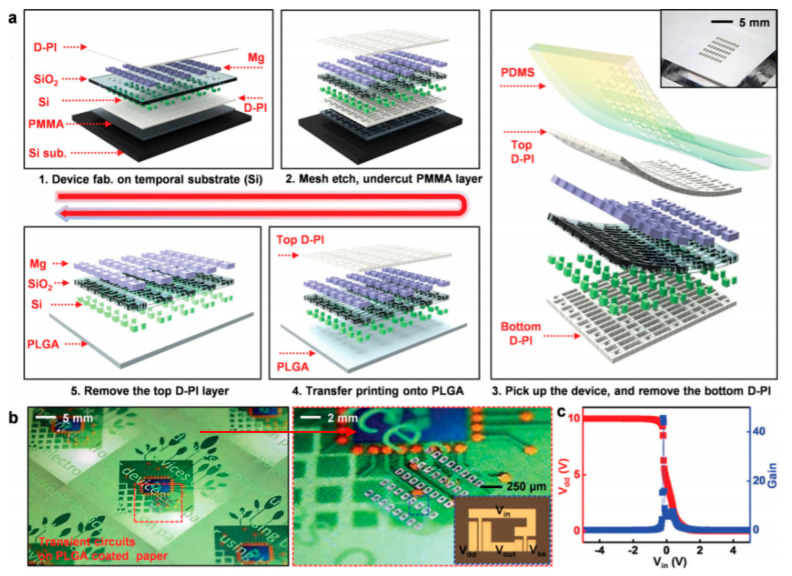
(**a**) Steps (1–5) used for the fabrication of transient electronic components on a carrier substrate. The layered structure allows dissolution of the sacrificial PMMA layer (on top of the Si substrate—1) in acetone to release the device (2). Collection of the released device onto a PDMS stamp (3), followed by transfer printing onto a PLGA substrate (4) then RIE (Reactive Ion Etching) of the top diluted polyimide (D-PI) (5). (**b**) (**left**) An array of transient CMOS inverters on PLGA, deposited onto a printed paper sheet (magnified view on the **right**), with a microscope image of one single inverter in the inset. (**c**) Output characteristics of an inverter obtained by this technique (V_dd_ = 10 V). Reproduced from [64] with permission. Copyright © 2014 Wiley and Sons.

**Figure 9 sensors-20-05898-f009:**
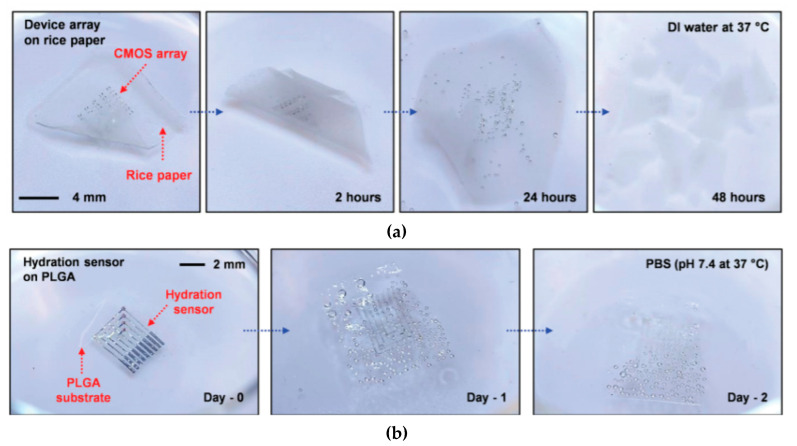
Optical images at various stages of dissolution of (**a**) an array of transient CMOS inverters on rice paper upon immersion in DI water at 37 °C and (**b**) of a hydration sensor on a PLGA film in PBS. Reproduced from [64] with permission. Copyright © 2014 Wiley and Sons.

**Figure 10 sensors-20-05898-f010:**
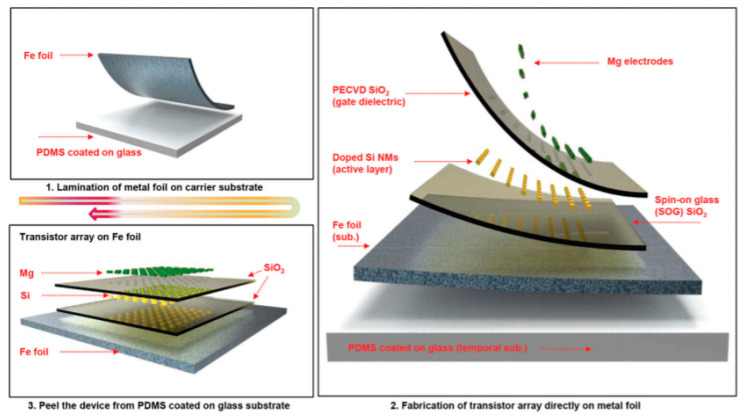
Fabrication steps of a transcient n-MOSFETs array on a biodegradable Fe foil: lamination of a Fe foil on a PDMS-coated glass substrate and exploded views of one MOSFET. Detachment from the PDMS/glass substrate gives a free-standing film. The materials include silicon nanomembranes (Si NMs; semiconductor), thin Mg films (conductor), spin-on glass (SOG) SiO_2_ (dielectric), and Fe foil (substrate). Adapted from [45] with permission. Copyright © 2015 Wiley and Sons.

**Figure 11 sensors-20-05898-f011:**
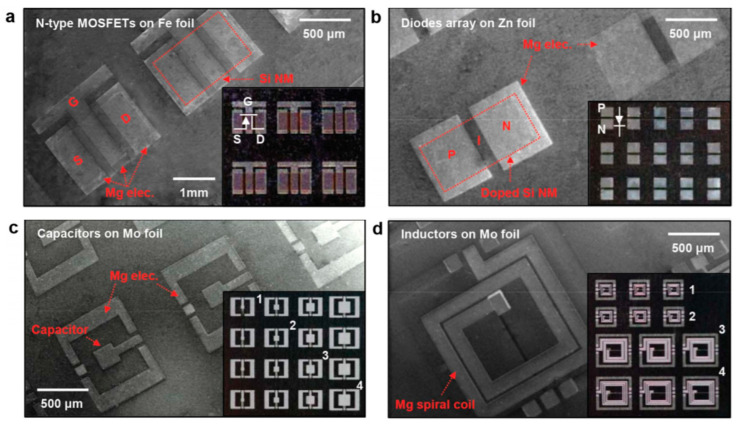
Scanning electron microscope (SEM) and optical microscope image (inset) of diverse transient devices on dissolvable metal foil substrates. (**a**) array of nMOSFETs on Fe foil. (**b**) Transient PN diode on a Zn foil. (**c**) Capacitors with different sizes built using Mg electrodes (top/bottom) and PECVD SiO_2_ dielectrics on Mo foil. (**d**) Planar spiral coils on a Mo foil substrate. Adapted from [45] with permission. Copyright © 2015 Wiley and Sons.

**Figure 12 sensors-20-05898-f012:**
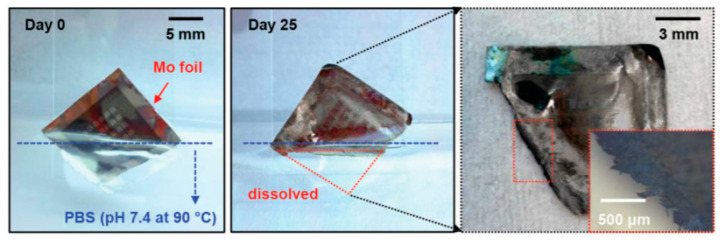
Dissolution of a transistor array on a Mo foil (5 µm thick) partially immersed in phosphate buffer solution (PBS, pH 7.4) at 90 °C. Device in its initial state (left) and partially dissolved after 25 days (middle). Magnified (right) and microscope (inset) images show full dissolution of the immersed area and partial dissolution at the edge. Addapted from [45] with permission. Copyright © 2015 Wiley and Sons.

**Figure 13 sensors-20-05898-f013:**
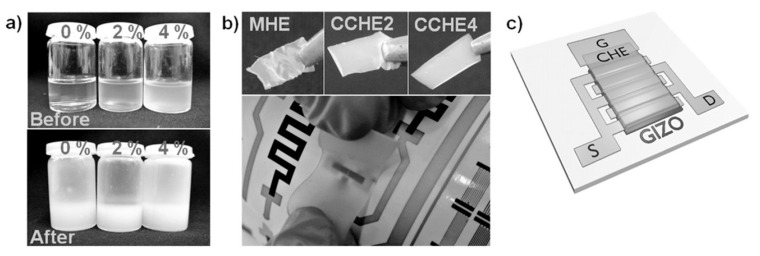
(**a**) Solutions containing various weight ratios of carboxymethyl cellulose (CMC) (0, 2 and 4 wt.%) before and after neutralization with acetic acid. (**b**) Resulting flexible and transparent free-standing cellulose-based hydrogel electrolytes (CHEs) obtained from the respective solutions (MHE, CCHE2, CCHE4). (**c**) Schematic illustration of a CHE-gated IGZO EGT. Reproduced from [70] with permission. Copyright © 2017 Wiley and Sons.

**Figure 14 sensors-20-05898-f014:**
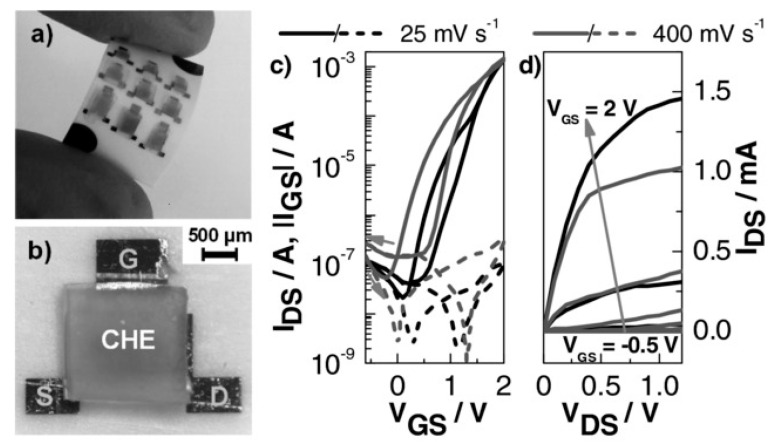
(**a**) Picture of an IGZO-EGT array on paper, with CCHE4 as a solid electrolyte. (**b**) Details of a single EGT. (**c**) Transfer characteristics at two different *V*_GS_ scan rates. Dashed lines correspond to *I*_GS_. (**d**) output curves (gate voltage step of 0.25 V, starting from −0.5 up to 2 V). Reproduced from [70] with permission. Copyright © 2017 Wiley and Sons.

**Figure 15 sensors-20-05898-f015:**
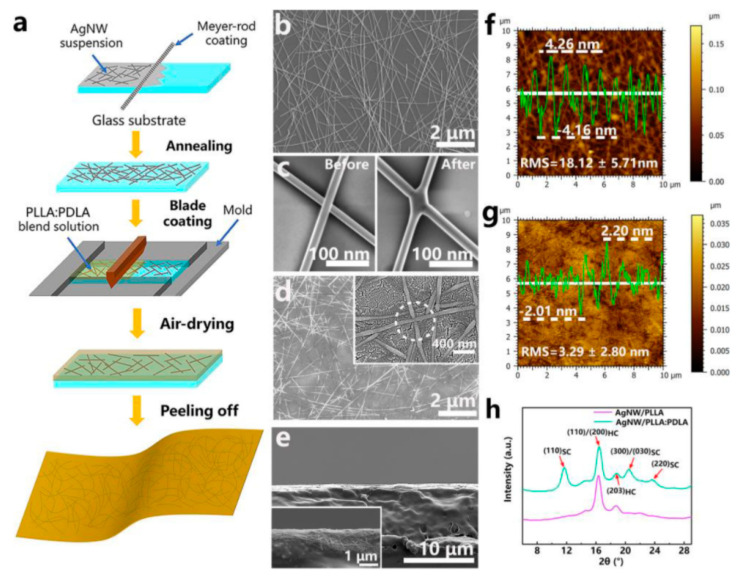
(**a**) Fabrication steps of aAgNWs/PLA film. SEM images of (**b**) the blended film + AgNWs. (**c**) Detail of AgNWs film before and after annealing. (**d**) The AgNW/PLLA:PDLA film after annealing. (**e**) Cross-sectional SEM view. AFM images of (**f**) the AgNW/Glass film and (**g**) the AgNWs/PLLA:PDLA film. (**h**) WAXD profiles of the AgNW/PLLA and AgNW/PLLA:PDLA films. Reproduced from [71] under a Creative Commons Creative Common CC BY License.

**Figure 16 sensors-20-05898-f016:**
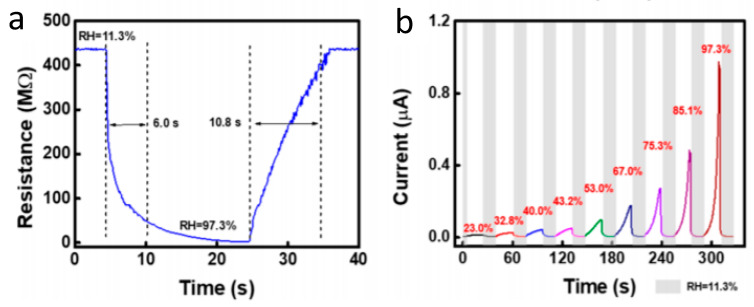
(**a**) Response/recovery times of cellulose/KOH film (CKF) for humidity levels between 11.3 and 97.3% RH. (**b**) Real-time current response of CKF to RH ranging from 11.3 to 97.3%. Adapted with permission from [75]. Copyright 2020 American Chemical Society.

**Figure 17 sensors-20-05898-f017:**
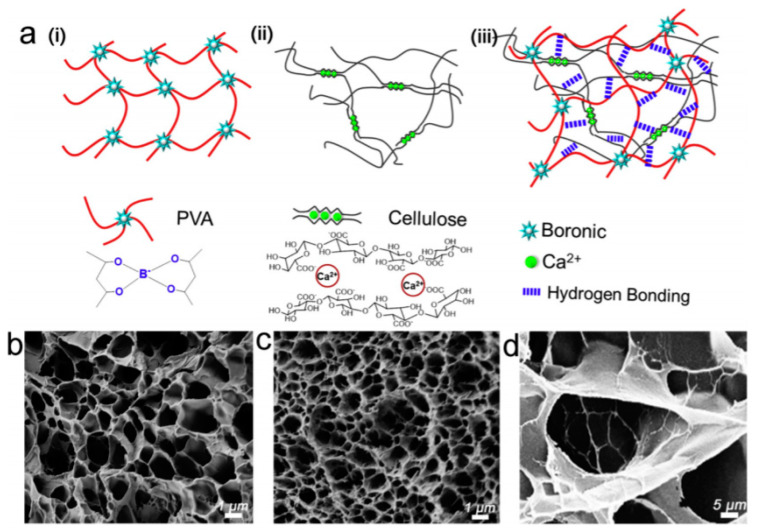
(**a**) Structure of the self-healing hydrogel: (**i**) PVA hydrogel, (**ii**) cellulose nanofiber (CNF) network, and (**iii**) PVA/CNF hydrogel bend. (**b**) Morphologies of freeze-dried PVA hydrogel. (**c**) and (**d**) Morphologies of the freeze-dried PVA/CNF hydrogel. Reproduced from [76] with permission. Copyright 2019, Elsevier.

**Figure 18 sensors-20-05898-f018:**
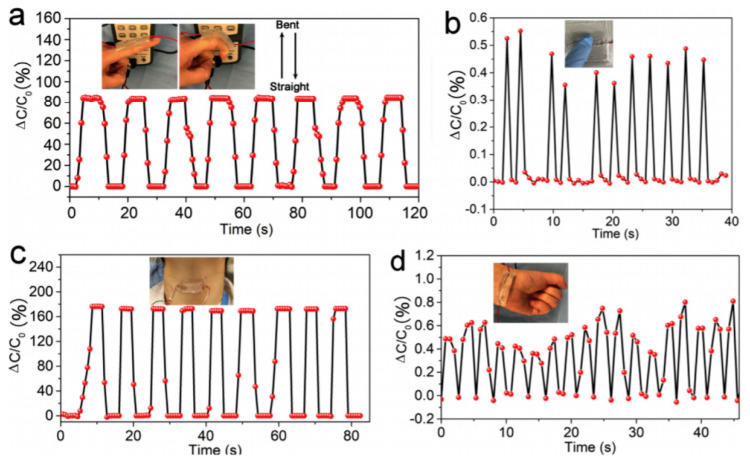
(**a**) Photographs of a hydrogel sensor attached on top of a forefinger and used to detect its motions. (**b**) Response of the hydrogel sensor upon a gentle touch with a finger. (**c**) Real-time capacitance change in the hydrogel sensor attached to the throat to detect deglutition. (**d**) Real-time capacitance signals from the hydrogel sensor when attached to the wrist to measure heartbeat. Reproduced from [76] with permission. Copyright 2019, Elsevier.

**Figure 19 sensors-20-05898-f019:**
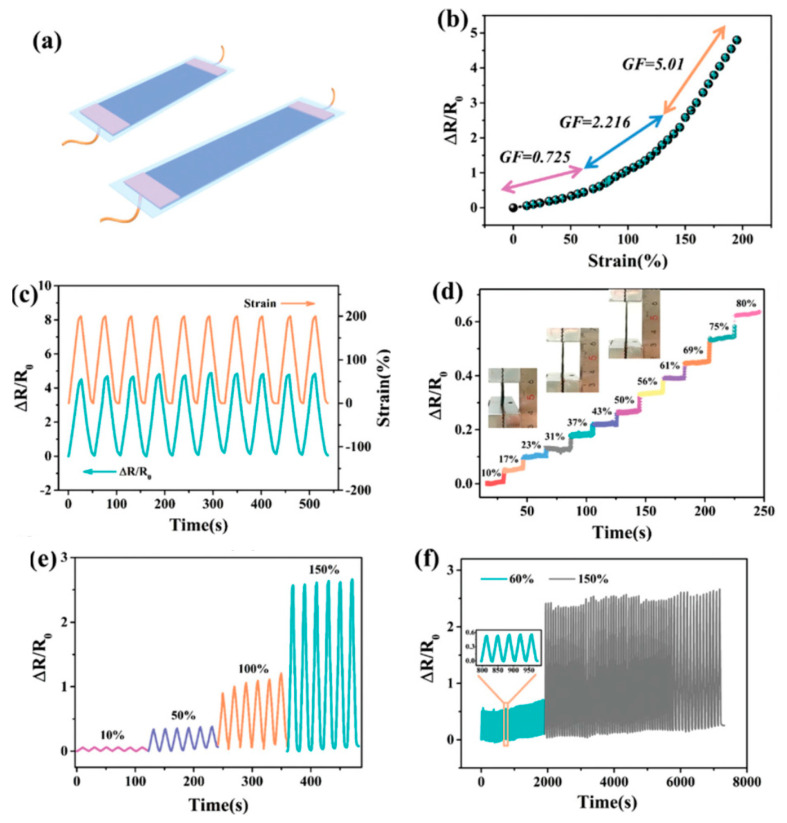
Proof-of-concept of the PVA/SA/BC/MCC hydrogel as piezoresistive strain sensor. (**a**) Schematic of the hydrogel sample upon stretching. (**b**) Relative resistance changes (RRC) as a function of the applied strain. (**c**) Plots of RRC and strain. (**d**) RRC for various tensile strains from 10% to 80% of its maximum. (**e**) RRC upon cyclic applied strains (10%, 50%, 100%, 150%). (**f**) The same, for a longer time. Reproduced from [81] with permission. Copyright © 2019 Wiley and Sons.

**Figure 20 sensors-20-05898-f020:**
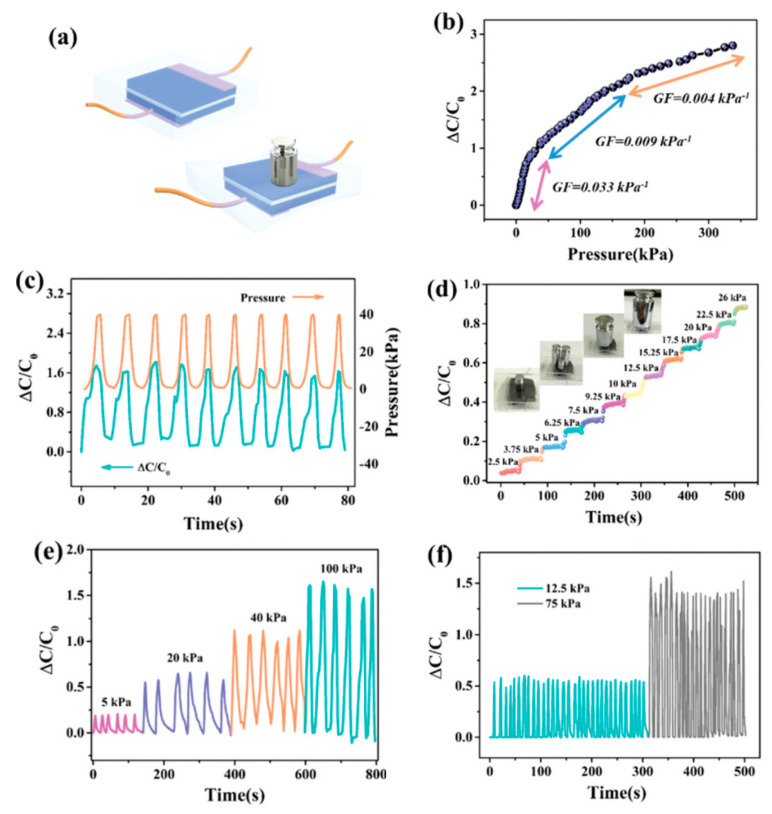
Proof-of-concept of the PVA/SA/BC/MCC hydrogels as a capacitive pressure sensor. (**a**) Schematic of the pressure sensor. (**b**) Relative capacitance change (RCC) as a function of the applied pressure. (**c**) RCC and pressure measured in parallel. (**d**) RCC under increasing applied pressures from 2.5 to 26 kPa. (**e**) RRC as a function of a cyclic applied pressure from 5 to 100 kPa. (**f**) RCC under compressing/releasing cycles. Reproduced from [81] with permission. Copyright © 2019 Wiley and Sons.

**Figure 21 sensors-20-05898-f021:**
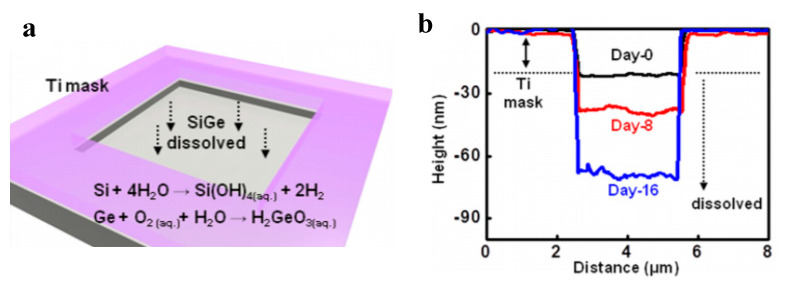
(**a**) Scheme of the experimental setup used to measure the Si, Ge and SiGe dissolution rate, in the form of a patterned square opening (3 μm × 3 μm × 30 nm) within a Ti layer deposited on single-crystalline I, SiGe and Ge (100) wafers. (**b**) Height profiles during hydrolysis of SiGe in a buffer solution at pH 10 and 37 °C, after day 0, day 8 and day 16. Adapted from from [91]. Copyright 2015 American Chemical Society.

**Figure 22 sensors-20-05898-f022:**
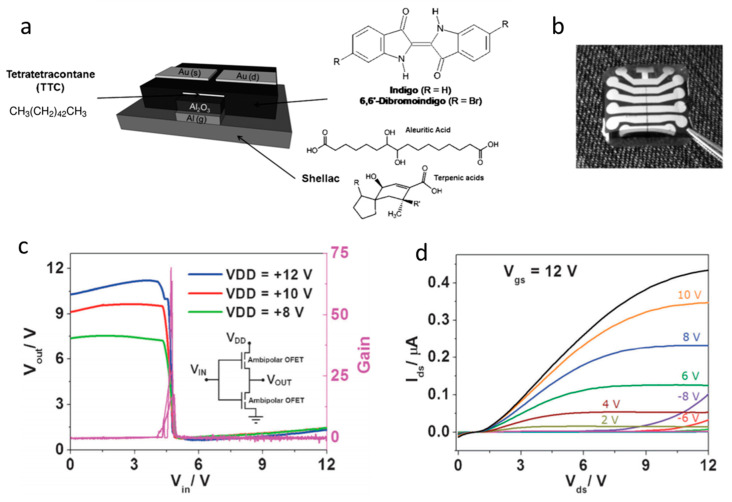
(**a**) The OFET device, using tetratretracontane (TTC) between the Al_2_O_3_ dielectric and the indigo (or dibromoindigo) semiconductor. The substrate is a shellac (varnish) layer. (**b**) Pictures of a five-transistor sample on a shellac substrate. The W and L parameters are 1 and 80 mm, respectively. The capacitance of the TTC/Al_2_O_3_ layer is was ca. 90 nF cm^−2^. (**c**) Quasi steady-state output characteristics of the indigo-based inverter. (**d**) Output characteristics. Adapted from [98]. Copyright © 2012 Wiley and Sons.

**Figure 23 sensors-20-05898-f023:**
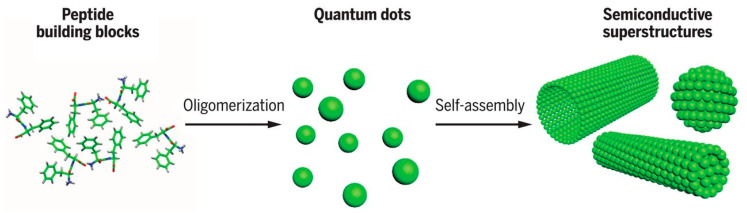
(**top**) Examples of peptide self-assembling architectures: peptide building blocks oligomerize to form quantum dots, themselves serving as building blocks which self-assemble into supramolecular structures with semiconducting properties. Reproduced from [99]. Copyright 2017, the American Association for the Advancement of Science. (**bottom**) Molecular mechanisms underlying short peptide self-assembling semiconductors. (**A**) Model showing diphenylalanine (FF) nanotubular crystals acting as quantum confined structures comprising a tubular backbone (red circle) surrounded by six FF units (cyan circle). Adapted from [100]. Copyright 2014 American Chemical Society. (**B**) Cross-sectionnal view of two adjacent peptide β-sheets. The grayed area points out a quantum confined region. Adapted from [101]. Copyright 2016 American Chemical Society. (**C**) Quantum confined crystals of FF and phenylalanine-tryptophan (FW), and their respective conduction and valence band. FW shows a smaller band gap (the conduction band is lowered). Adapted from [102]. Copyright 2015 AIP Publishing.

**Figure 24 sensors-20-05898-f024:**
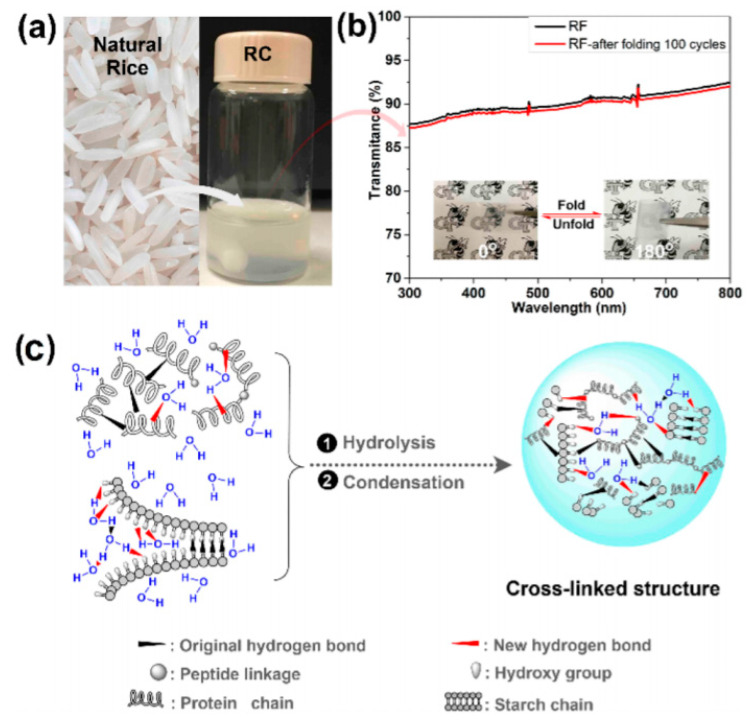
(**a**) Photographs of the rice coloidal solution (RC). (**b**) Transmittance of the free-standing rice film (RF) before and after a sequence of 100 foldings at 180°. (**c**) Structures and mechanism for the formation of the cross-linked structure within the rice film, which provides self-standing properties. Reproduced from [106]. Copyright 2016 American Chemical Society.

**Figure 25 sensors-20-05898-f025:**
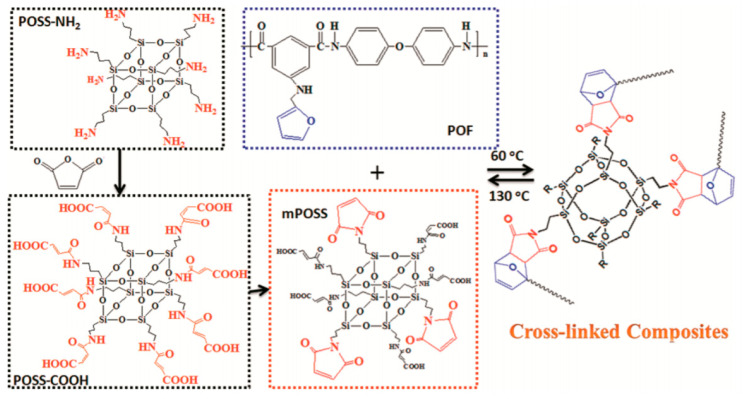
Synthetic route to obtain the cross-linked polyhedral oligomeric silsesquioxane (POSS) and thermal reversibility. Reproduced from [114]. Copyright 2016 American Chemical Society.

**Figure 26 sensors-20-05898-f026:**
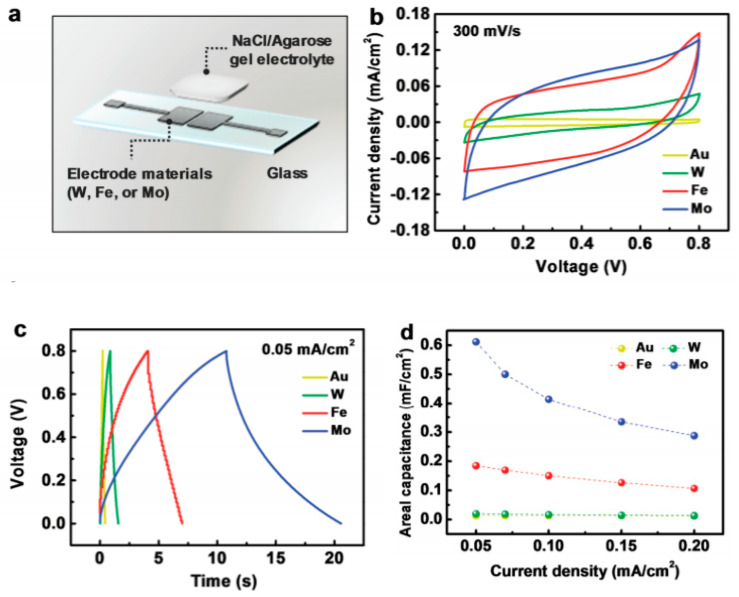
(**a**) Scheme of a supercapacitor made of biodegradable metal electrodes (W, Fe, or Mo) and an agarose gel electrolyte. (**b**) and (**c**) Electrochemical characterizations. (**d**) Capacitance as a function of current density. (**e**) Charge/discharge curves (normalized capacitance). (**f**) Impedance (Nyquist) plots of the supercapacitor before and after 500 charge/discharge cycles. Reproduced from [119]. Copyright © 2017 Wiley and Sons.

**Figure 27 sensors-20-05898-f027:**
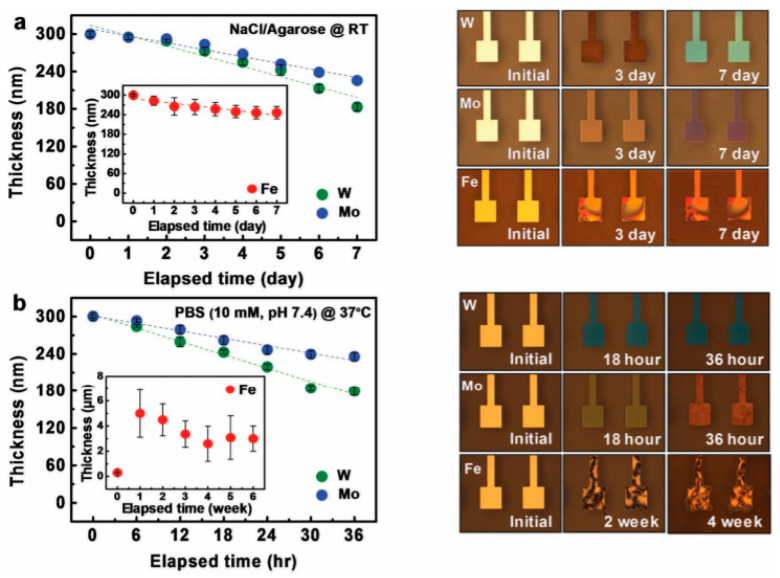
(**Left**) Dissolution kinetics (**a**) at room temperature and (**b**) at 37 °C of the metal (W, Mo and Fe) thin films, in phosphate-buffered saline and with the agarose gel on top. (**Right**) optical images at several times. Adapted from [119]. Copyright © 2017 Wiley and Sons.

**Figure 28 sensors-20-05898-f028:**
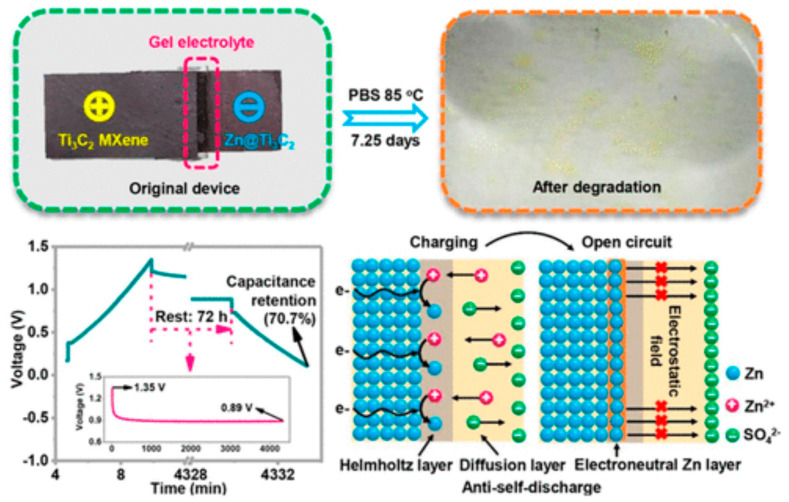
(**Top**) Anode and cathode of the gel capacitor device, with Ti_3_C_2_ as anode and Zn@Ti_3_C_2_ as cathode, before then after (**right**) degradatio in PBS at 85 °C during a week. (**Bottom**) Charge–discharge curves (**left**) and schematized functioning (**right**). Reproduced from [120]. Copyright 2019 American Chemical Society.

**Figure 29 sensors-20-05898-f029:**
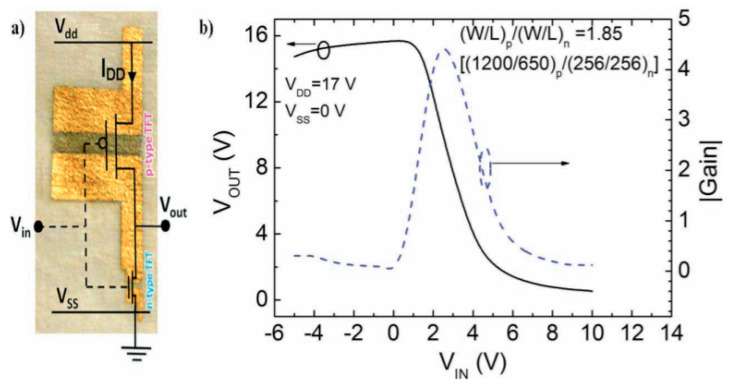
(**a**) Picture of a paper CMOS inverter with a p-FET of W/L = 22 a n-FET of W/L = 12. (**b**) Electrical characteristics of the inverter. Reproduced from [46] with permission. Copyright © 2012 Wiley and Sons.

**Figure 30 sensors-20-05898-f030:**
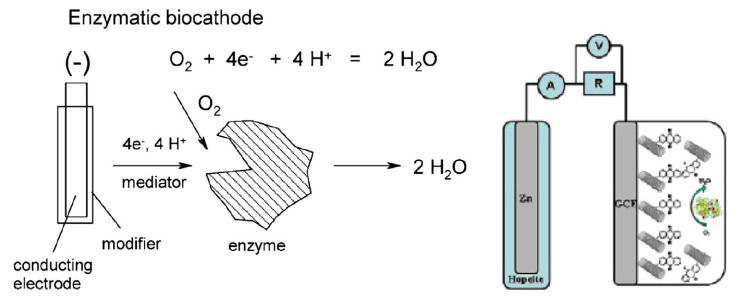
Oxygen reduction on an enzymatic biocathode, and scheme of the complete biobattery, comprising a Zn anode onto which a hydrated zinc phosphate layer was grown, and a glassy carbon cathode modified by biphenylated SWCNTs and an enzyme. Reproduced from [139] with permission. Copyright 2013 from Elsevier.

**Figure 31 sensors-20-05898-f031:**
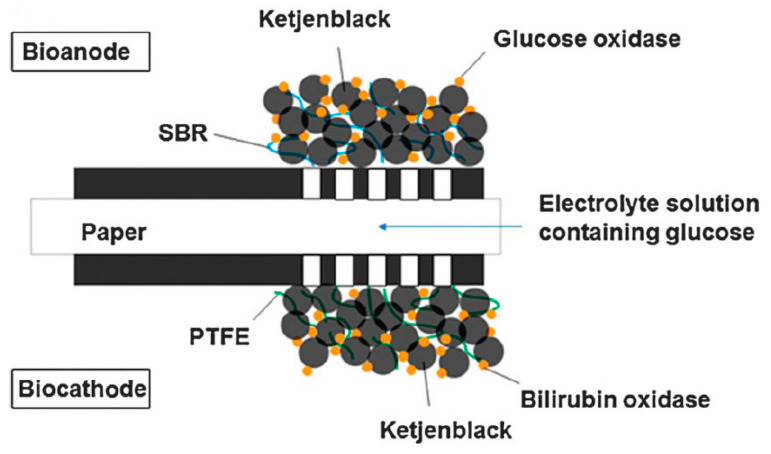
Sceen-printed, paper-based carbon electrodes as bioanodes and biocathodes for a glucose oxidase/bilirubin oxidase BFC. SBR: styrene butadiene rubber. PTFE: polytetrafluoroethylene. Ketjenblack: superconducting carbon black. Reproduced from [145] with permission. Copyright The Royal Society of Chemistry, 2013.

**Figure 32 sensors-20-05898-f032:**
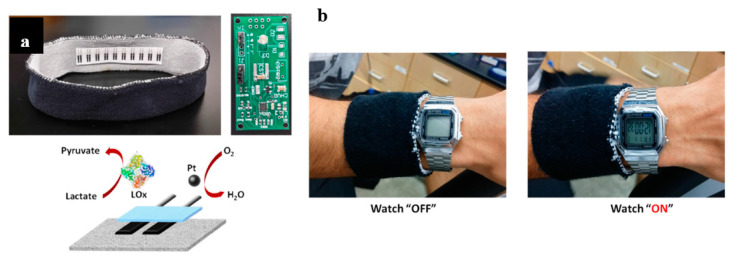
(**a**) The wristband biofuel cell, its DC/DC converter and illustration of the reaction taking place at the two electrodes. (**b**) wristwatch powered by human sweat. Adapted with permissionfrom [150]. Copyright 2016 WILEY-VCH Verlag GmbH & Co. KGaA, Weinheim.

**Figure 33 sensors-20-05898-f033:**
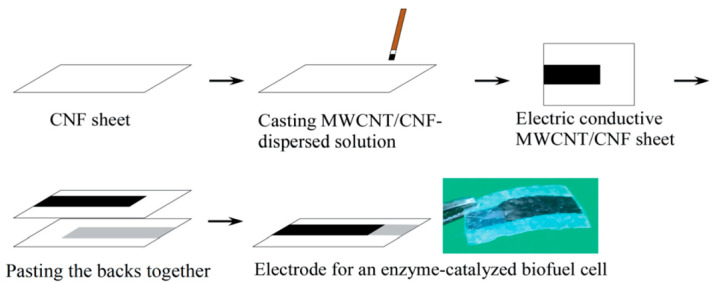
Illustration of the cellulose nanofiber (CNF)-based electrode in an enzymatic BFC. Reproduced from [152] under Creative Commons Attribution Non-Commercial 3.0 Unported Licence.

**Figure 34 sensors-20-05898-f034:**
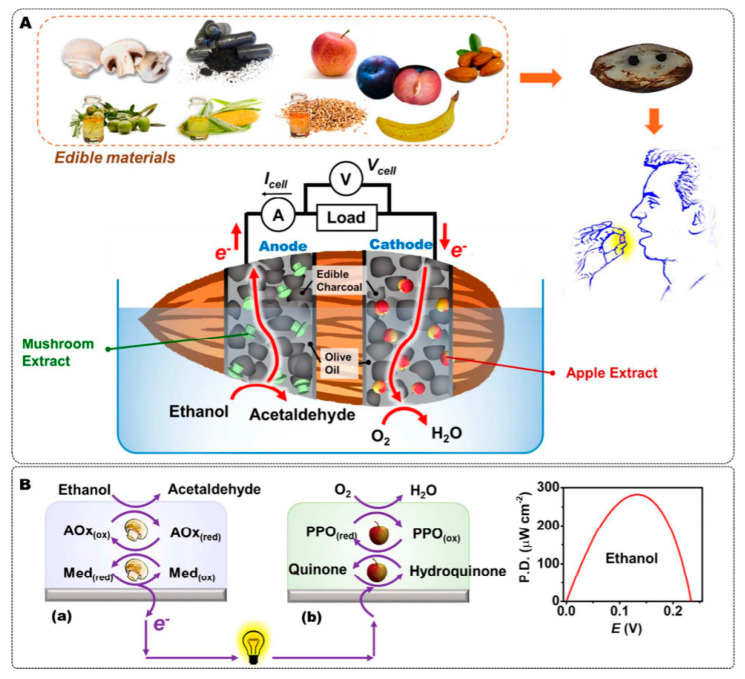
(**A**) Schematic of the edible ethanol biofuel cell serving as ethanol sensor, into which all components come from edible products. (**B**) Reactions occuring at the anodic and cathodic compartments. AOx is for alcohol oxidase, while PPO is for polyphenol oxidase. The maximum power developped by this device is ca. 280 μW cm^−2^ at open circuit. Reproduced from [153] with permission. Copyright 2018 Royal Society of Chemistry.

**Figure 35 sensors-20-05898-f035:**
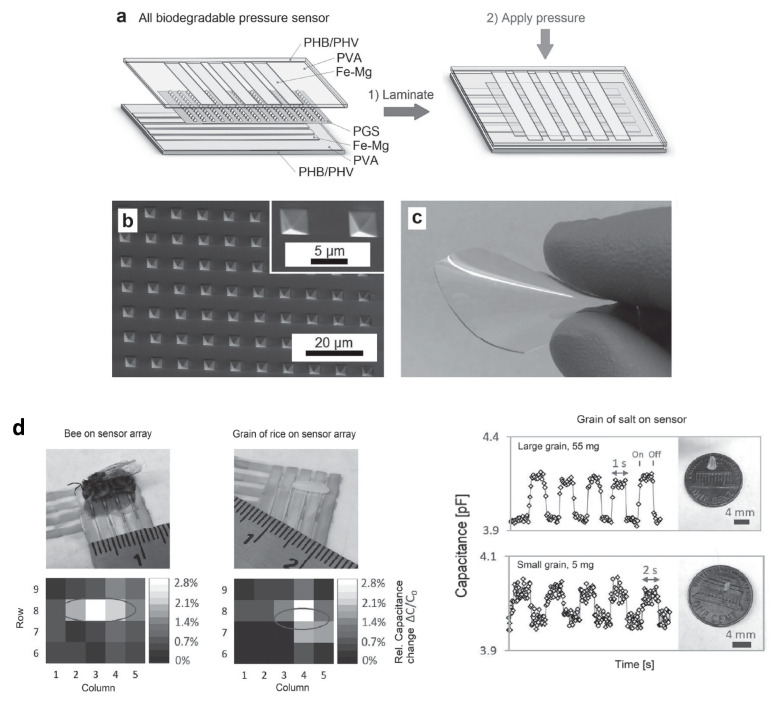
(**a**) Fabrication procedure of the fully biodegradable and flexible pressure sensor array from microstructured poly(glycerol sebacate) (PGS), polyhydroxybutyrate/polyhydroxyvalerate (PHB/PHV) and Fe/Mg as conductors. (**b**) SEM images of the microstructured PGS films (PGS was formed into a PDMS mold). (**c**) Flexible PGS film. (**d**) (**left**) Picture of the sensor array of 4 × 5 pressure-sensitive elements and capacitance change distribution on the sensor array upon placing light weights such as a bee (22.5 mg), and a grain of rice (21.8 mg). (**right**) capacitance changes as a function of the weight of the object. Adapted from [156] with permission. Copyright © 2015 Wiley and Sons.

**Figure 36 sensors-20-05898-f036:**
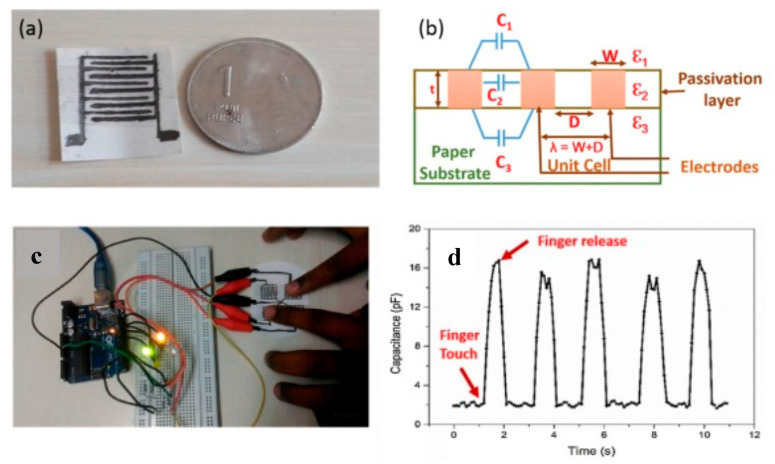
(**a**) Picture of the interdigitated capacitive sensor, with tracks made remarkably simply with a graphite pencil on paper. (**b**) Electrical details of the sensor and its capacitances. (**c**) Photograph of the functioning touchpad. (**d**) Variation in capacitance of a single sensor upon finger contact. Adapted from [159] with permission. Copyright 2017 from Elsevier.

**Figure 37 sensors-20-05898-f037:**
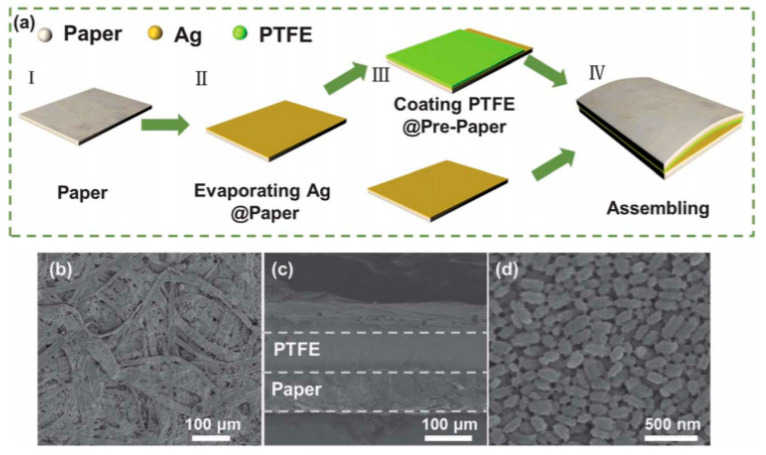
(**a**) Scheme illustrating the fabrication process of the paper generator. (**b—d**) SEM images of the paper coated with an Ag layer (**left**), the cross-section of the Ag-paper covered by PTFE (**middle**) and the top view of the Ag–paper after spin-coating with PTFE. Reproduced from [160]. Copyright 2013, The Royal Society of Chemistry.

**Figure 38 sensors-20-05898-f038:**
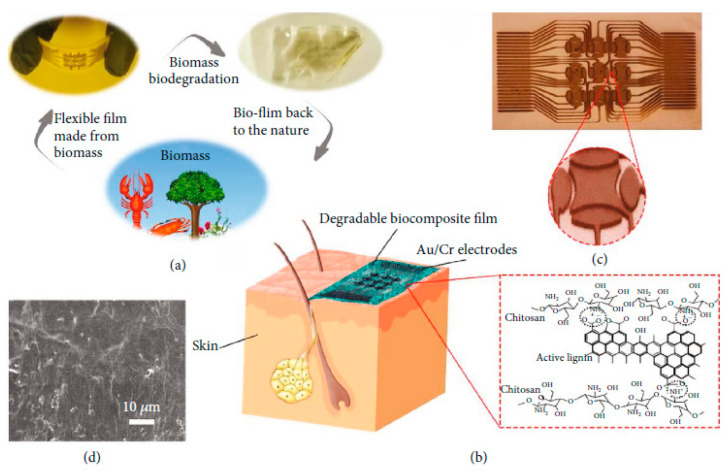
(**a**) Scheme of the life cycle for the degradable and natural sensor made of chitin, chitosan and lignin. (**b**) Details of the humidity sensor on skin, with the chemical structures of its constituents. (**c**) Micrograph of the electronic structure of the device. (**d**) SEM image of functionalized polysaccharide film. Reproduced from [161] with permission. Copyright © 2020 L. Wang et al. Exclusive Licensee Science and Technology Review Publishing House. Distributed under a Creative Commons Attribution License (CC BY 4.0).

**Figure 39 sensors-20-05898-f039:**
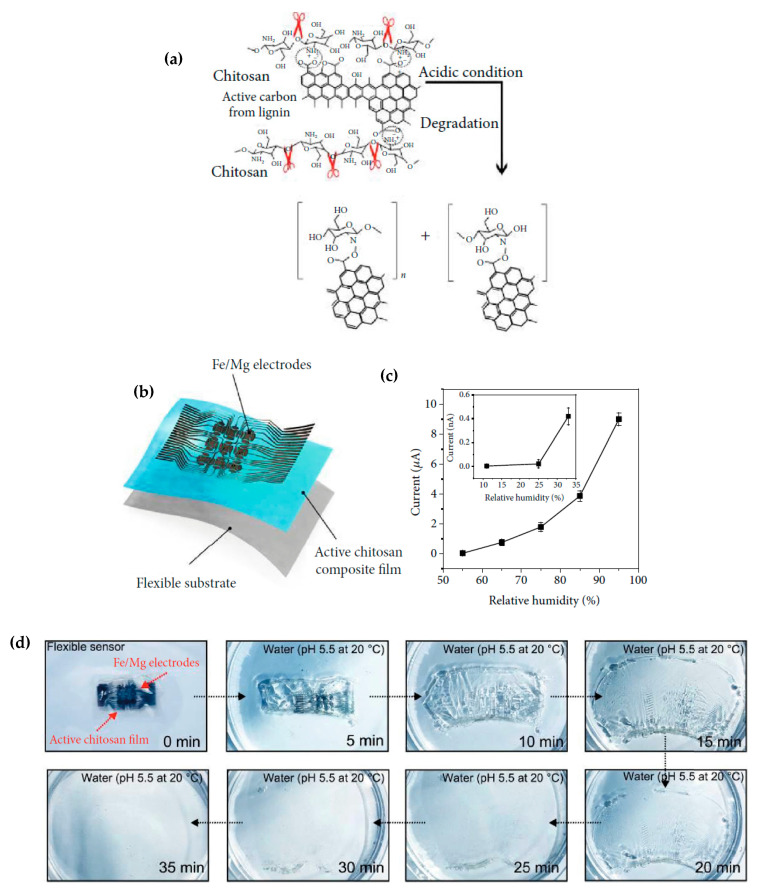
(**a**) Mechanism for the preparation of the biodegradable substrate. (**b**) Illustration of the flexible device. (**c**) Changes in current as a function of humidity. (**d**) Pictures of the progressive degradation upon insertion into water (pH 5.5) at room temperature. (**e**) Humidity-sensing properties: current under different RH (voltage bias of 1 V); Relative current changes under increasing %RH; Dynamic current change under 85%RH for 20s; Relative current changes upon repetitive exposure to 95%RH. Adapted from [161] with permission. Copyright © 2020 L. Wang et al. Exclusive Licensee Science and Technology Review Publishing House. Distributed under a Creative Commons Attribution License (CC BY 4.0).

**Figure 40 sensors-20-05898-f040:**
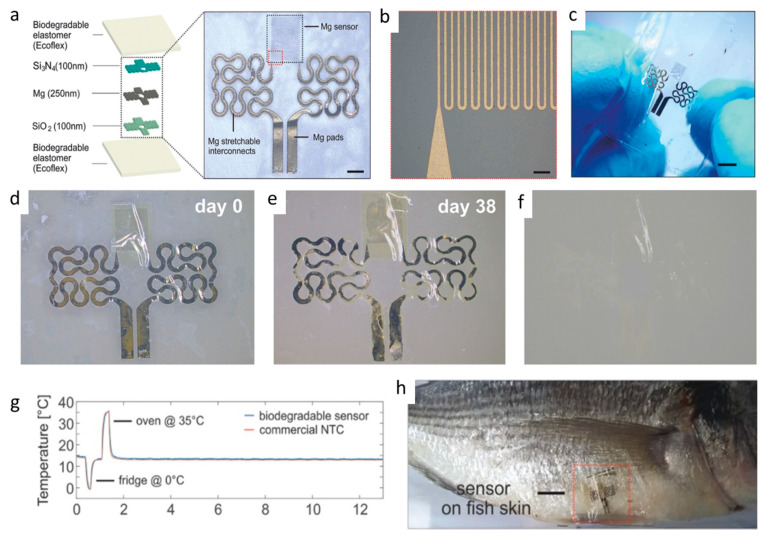
(**a**) The sensor consists of Mg serpentine, interconnections and contact pads. Si_3_N_4_, SiO_2_ and Ecoflex are used as encapsulation layers. (**b**) Optical micrograph or a part of the serpentine sensing element (scale bar 50 µm). (**c**) Picture of the whole biodegradable sensors (scale bar 1 mm). Image of the sensor (**d**) before immersion in water–NaCl solution (150 mmol) at 25 °C, then (**e**) after 36 days and (**f**) after 67 days. (**g**) Temperature measurement using the biodegradable sensor, compared with a conventional negative temperature coefficient (NTC) thermistor. (**h**) Example of use: the sensor is stick on a fish to track shipping and storage temperatures. Adapted from [162] with permission. © 2017 WILEY-VCH Verlag GmbH & Co. KGaA, Weinheim

**Figure 41 sensors-20-05898-f041:**
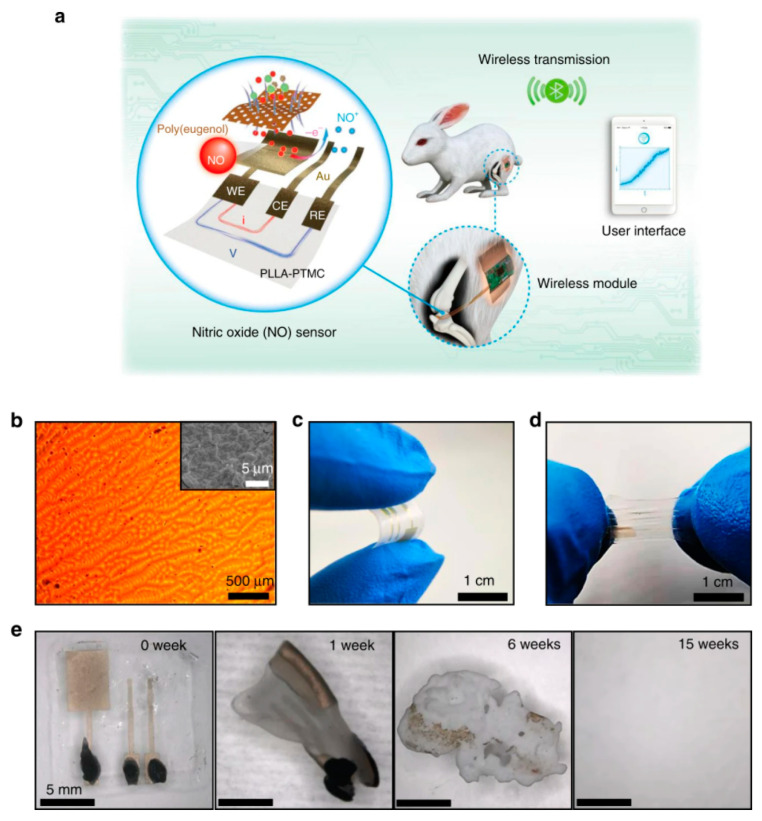
(**a**) Illustration of the bioresorbable NO sensor. The sensor is made of Au electrodes, poly(eugenol) and poly(L-lactic acid)–poly(trimethylene carbonate) (PLLA–PTMC) as substrate. NO concentration is measured by amperometry. The sensor is implanted in the joint cavity of rabbit. (**b**) Optical and SEM images of the surface morphology of Au electrodes bearing a poly(eugenol) film. (**c**,**d**) Pictures of the NO sensor upon bending and stretching. (**e**) Pictures of the sensor acquired during accelerated degradation in PBS at 65 °C. Adapted from [163], licensed under a Creative Commons Attribution 4.0 International License.

**Figure 42 sensors-20-05898-f042:**
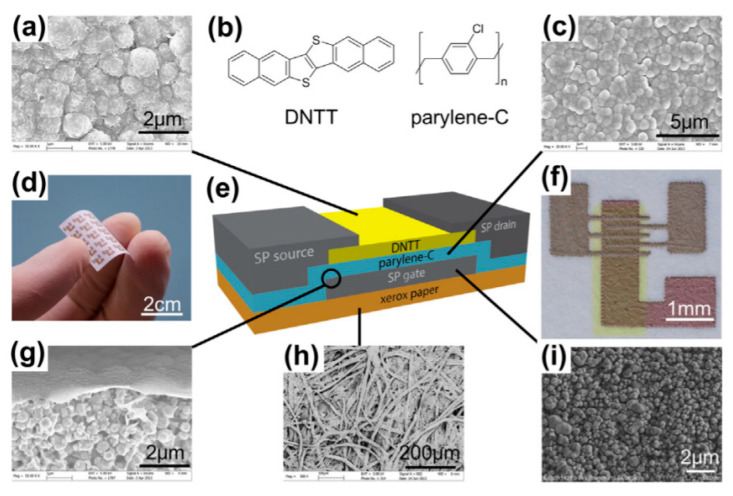
(**a**) SEM image of the semiconducting DNTT (dinaphthothienothiophene) surface. (**b**) Chemical structure of DNTT and of the parylene-C dielectric. (**c**) SEM image of parylene-C surface. (**d**) Picture of 5 × 5 transistor matrix on Xerox paper. (**e**) Scheme of the whole transistor on paper. Source, drain and gate are scree-printed. (**f**) Picture of a single transistor on paper. (**g**) SEM cross-view of the parylene-C layer on top of the gate. (**h**) SEM picture of the paper surface. (**i**) SEM image of the gate surface. Adapted from [169], copyright 2014, with permission from Elsevier.

**Figure 43 sensors-20-05898-f043:**
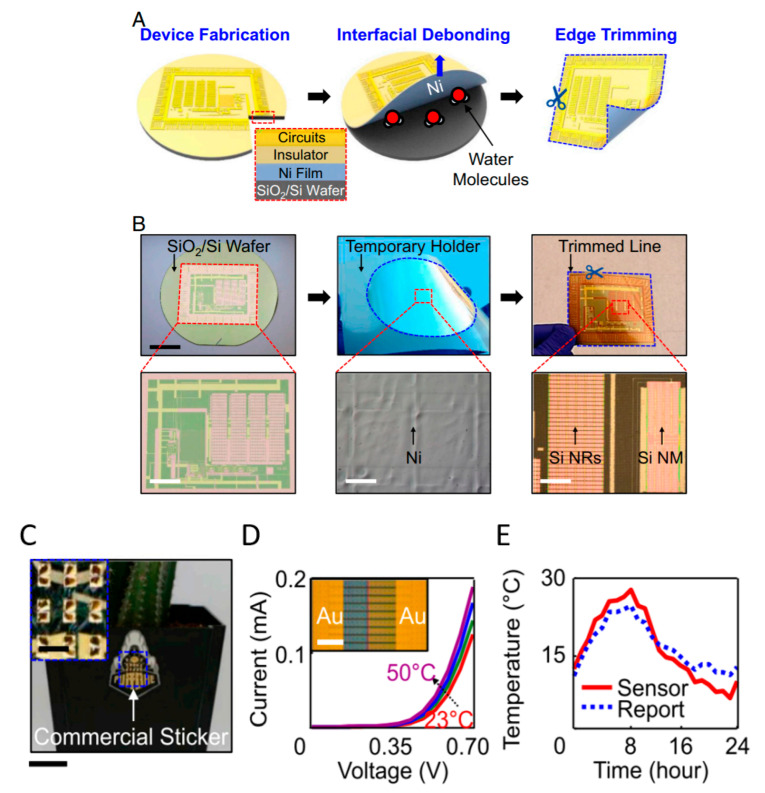
(**A**) Key steps of the fabrication and debonding of thin-film nanoelectronics from conventional Si wafers; Inset: cross-sectional view of the successive layers. (**B**) Optical images of the thin films: (left), on the wafer (scale bar: 2.5 cm); (middle), peeled with a thermally releasable tape; (right) trimmed. (**C**) Optical image of arrays of a Si -based temperature sensor pasted on the surface of a cactus pot (scale bar: 2 cm). Inset: magnified view of the sensor sticker. (**D**) Corresponding electrical characteristics for a temperature ranging from 23 to 50 °C. Inset: micrograph of the sensor (scale bar: 150 μm). (**E**) Environmental temperature measured by the above sensor, compared with a local weather report over a period of 24 h. Adapted from [173] with permission.

**Figure 44 sensors-20-05898-f044:**
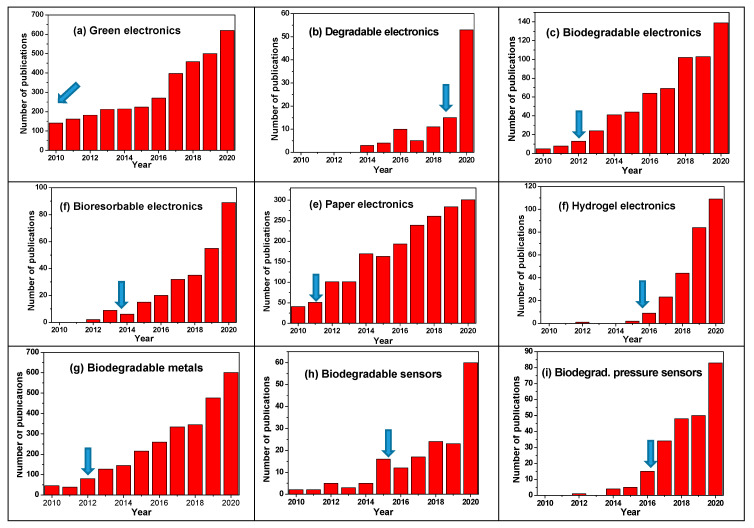
Histograms showing the progress, in terms of publication numbers which use (**a**) green electronics, (**b**) degradable electronics, (**c**) biodegradable electronics, (**d**) bioresorbable electronics, (**e**) paper electronics, (**f**) hydrogel electronics, (**g**) biodegradable metals, (**h**) biodegradable sensors and (**i**) biodegradable pressure sensors. The blue arrow indicates when the publication rate starts to increase significantly.

**Table 1 sensors-20-05898-t001:** Glossary of acronyms used in this review, sorted by alphabetical order.

Acronyms	Definitions	Acronyms	Definitions
AFM	Atomic Force Microscopy	RIE	Reactive-Ion Etching
BFC	Biofuel Cell	PBS	Phosphate Buffer Saline
CCR	Carbon Composition Resistance	PCB	Printed Circuit Board
CMOS	Complementary Metal Oxide Semiconductor	PDMS	Poly(DimethylSiloxane)
DNTT	DiNaphtho[2,3-b:2′,3′-f]Thieno[3,2-b]Thiophene	PEDOT	Poly(3,4-ethylenedioxythiophene)
DPPDTT	Poly(3,6-di (2-thien-5-yl)-2,5-di (2-octyldodecyl)-Pyrrolo [3,4-c] Pyrrole-1,4-Dione)Thieno [3,2-b] Thiophene)	PECVD	Plasma-Enhanced Chemical Vapor Deposition
EDLC	Electrochemical Double Layer Capacitor	P3HT	Poly(3-hexylthiophene)
EGT	Electrolyte-Gated Transistor	PI	Poly(imide)
FET	Field-Effect Transistor	PMMA	Poly(MethylMethAcrylate)
HBT	Heterojunction Bipolar Transistors	PSS	Poly(styrene sulfonate)
IoT	Internet of Things	PTCDI-C8	N,N′-Dioctyl-3,4,9,10-perylenedicarboximide
LED	Light Emitting Diode	PTFE	PolyTetraFluoroEthylene
MOSFET	Metal Oxide Silicon Field-Effect Transistor	RRC	Relative Resistance Changes
NTA	NanoTube Array	SEM	Scanning Electron Microscopy
NTC	Negative Temperature Coefficient	TCR	Temperature Coefficient Resistance
OFET	Organic Field-Effect Transistor		

**Table 2 sensors-20-05898-t002:** Different kind of (bio)degradable materials currently used in green electronics.

Name	Structure	Full Name
Ag	-	Silver
AgNW	-	Silver NanoWire
Al	-	Aluminum
CHECKFCNF	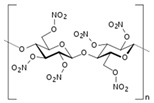	Cellulose-based Hydrogel ElectrolytesCellulose/KOH FilmCellulose NanoFibril
DNA	-	DesoxyriboNucleic Acid
Fe	-	Iron
FF	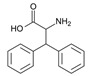	Diphenylalanine
FW	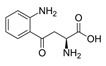	Phenylalanine-Triptophan
GO	-	Graphene Oxide
IGZO	-	Indium-Gallium-Zinc Oxide
Indigo	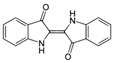	Indigo
IZO	-	Indium-Zinc Oxide
LCF	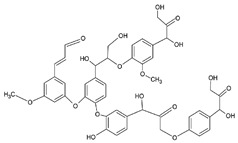	Lignin-derived Carbonized Nanofibers
Mg	-	Magnesium
	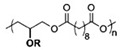	
PGS	W with R = H	Poly(Glycerol Sebacate)

PHB	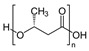	Poly(HydroxyButyrate)
PHV	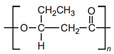	Poly(HydroxyValerate)
PLLAPDLAPLA	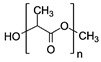	Poly(L-Lactic Acid)Poly(D-Lactic Acid)Poly(Lactic acid)
PLGA	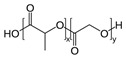	Poly(lactic-co-glycolic acid)
POMaC	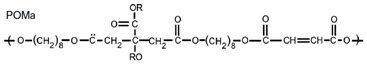	Poly(Octamethylene Maleate Anhydride Citrate)
POSS	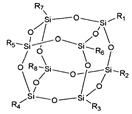	Polyhedral Oligomeric SilSesquioxane
PTMC	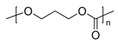	Poly(TriMethylene Carbonate)
PVA	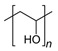	Poly(VinylAlcohol)
RF	-	Rice Film
rGO	-	Reduced Graphene Oxide
SA	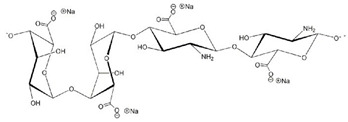	Sodium Alginate
Si	-	Silicon
SOG	-	Silica spin-On-Glass
TTC	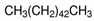	TetraTretraContane
Zn	-	Zinc

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
