# Peer review of "Sensors Made of Natural Renewable Materials: Efficiency, Recyclability or Biodegradability—The Green Electronics"

_sensors, 2020, doi:10.3390/s20205898_

Round 1

Reviewer 1 Report

Manuscript ID: sensors-929868

I recommend the acceptance of the manuscript after ‘minor revision’.

In this review paper entitled “Sensors Made of Natural Renewable Materials: Efficiency, Recyclability or Biodegradability. The Green Electronics.”,at first the authors have summarized on the use of different organic and inorganic passive materials as substrates, supporting matrixes or packaging in various sensors. Then they have reviewed on the use of different active materials (conductors or semiconductors)in different sensors.Overall, the manuscript is well organized and fits the journal´s scope, it is of interest. A lot of historical evidance on this topic is presented. I recommend its publication in “Sensors” after minor revision. Here are some comments:

  1. The introduction section of the manuscript is too long. Author should try to reduce the length of the introduction section. Too many background knowledge in the introduction part. It is better to brief the work in discussion section.
  2. There are some grammatical mistakes in the manuscript that must be revised. Some examples are given below:
  3. Page 5, line 143: “a feature which have……”
  4. Page 6, line 158: “in a first section….……”
  5. Page 7, line 198: “biosourced….……”
  6. Page 7, line 202: “For example….……paper”
  7. Page 7, line 202: “For example….……paper”
  8. Page 7, line 203: “Electron….……respectively”

And many more. Author should check the whole manuscript carefully.

  1. The conclusion is weak and should be improved. The authors should have an outlook review or comments on the application of various natural renewable materials in sensors in the near future.
  2. The font size of the labels of some figures is very small. Therefore, it is very hard for readers to recognize them. Authors should resolve this problem in revised manuscript.

Author Response

We thank the reviewer for her/his constructive comments.

We reshaped the manuscript by moving parts of the introduction section into other sections. A section “Expected outcomes and definitions” has been created at the beginning of Section 2.

Many typos, orthographic and grammatical mistakes were found indeed, and corrected.

The concerned (too small) figures were resized to make the labels more readable, or the labels remade.

The conclusion has been strengthened following the comment, and conclusive remarks added at the end of several sections within the text.

Reviewer 2 Report

The authors present a review of sensors fabricated using renewable materials. This is quite an interesting review and will help the researchers to think more about sustainability and its impact on society. 

The manuscript however needs to be updated and would require important sections that have not been covered.

  1. The manuscript doesn't cover biofuels which have gained a lot of attention for green fuel and diagnostic applications.
  2. The authors should also have an outlook section where they should present their perspectives about the future of the field. What are the current limitations? It is about scalability? reliability? and so forth

Author Response

We thank the reviewer for her/his suggestion to add biofuel cells in the discussion. We now add a quite large section dealing with (bio)degradable biofuel cells.

Also, outlooks are now added at the end of several sections or subsections, and in the conclusion. 

Reviewer 3 Report

The topic of the paper is very interesting and related to the topics of the journal. I want to congratulate the authors for their efforts carried out in the search and analysis of the 157 references. Nonetheless, the paper must be heavily improved. Currently, it is only a compilation of work and figures of other authors, as an extremely long related work section. A proper survey must be more than this. It must be concise and direct to the point. Authors must include a new section describing the methodology followed to collect and analyze the papers. Include the selected keywords, the used search engines, if they have included any restriction to the search (by years or by topics). Authors must show more information, and not only the summary of each paper, it can be done in a new section before dealing with the content of papers themselves. For example, they can analyze the number of papers per year to determine if this topic is growing or not. The main countries that contribute to the topic and analyze why (if there is a reason) could also be detailed. This way, the authors will generate their own figures. They have to think if papers can be grouped by some characteristics, for example, area of application, type of material, etc…to compare and generate more graphics. Otherwise, a paper of 49  pages without any graphic or table that helps to summarize information is almost impossible to read by anyone.  Finally, and the most important issue for me, is that authors must include a section after analyzing the content of the included paper detailing the current challenges and future trends. In this section, authors should analyze the limitations on the existing technologies and draw possible future research lines and future topics based on their understanding of the current scenario after analyzing all the included papers in their survey.

Author Response

We thank the reviewer for his/her constructive remarks.

Concerning the methodology followed to collect and analyze papers, we added a comment in the last paragraph of the introduction section. This work is not supposed to be a comprehensive review collecting several hundreds of references, the domain being recent and not so rich in published examples. Therefore, a large majority of the cited publications are recent ones (2016-2020), the oldest having been published in 2012. We did not apply a systematic methodology but selected some works which attracted our interest.

As suggested, we discussed in the conclusion section about the growing interest of (bio)degradable electronics in general and sensors in particular and tried to identify the topics which are becoming the most attractive. We also identified the current challenges.